# MR–HCP: Morphology-Regularized Hierarchical Conformal Prediction for TEM of Subcellular Ultrastructure

## Abstract

Reliable uncertainty quantification is critical for deploying deep learning models in biomedical imaging, where fine-grained structures often exhibit overlapping morphology and ambiguous boundaries. We introduce *Morphology-Regularized Hierarchical Conformal Prediction (MR–HCP)*, a novel framework that combines hierarchical taxonomies with morphology-aware nonconformity to provide compact prediction sets with rigorous coverage guarantees. Unlike existing conformal methods that operate on flat label spaces and probability-only scores, our approach uses a two-stage super→fine calibration and penalizes morphological deviations from class prototypes. We integrate MR–HCP with a YOLOv11 detector to demonstrate an end-to-end pipeline for single-cell TEM analysis. On a curated neutrophil ultrastructure dataset (around 4.9k annotations), the method achieves near-nominal coverage (0.937), small average set size (1.12), and high singleton accuracy (0.932), significantly outperforming Split CP, Mondrian CP, HCC, and APS baselines. On the public Raabin–WBC dataset, it similarly attains strong coverage–set-size trade-offs compared to conformal baselines. Beyond inference, MR–HCP facilitates semi-automatic annotation and dataset reclassification, systematically refining coarse expert labels into fine categories while transparently quantifying uncertainty. Overall, the framework establishes a morphology-aware hierarchical conformal approach for uncertainty-aware classification in biomedical microscopy and provide a principled basis for extending calibrated set prediction to other hierarchical, morphology-rich biomedical settings.

## 1 Introduction

Transmission electron microscopy (TEM) enables subcellular analysis at nanometer scales, but fine-grained ultrastructure classification remains difficult: categories overlap in shape and intensity, boundaries are ambiguous, and expert labeling is time-consuming and inconsistent (Helmstaedter, 2013; Conrad & Narayan, 2023; Liu et al., 2022; Unnersjö-Jess et al., 2023). Standard deep models provide point predictions whose raw confidences are often miscalibrated under dataset shift, risking overconfident errors in scientific workflows (Horwath et al., 2020; Liu et al., 2025).

Conformal prediction (CP) offers finite-sample, distribution-free guarantees by returning a *set* of plausible labels that contains the truth with user-chosen probability (Vovk et al., 2005; Lei et al., 2018; Angelopoulos & Bates, 2021). This is well-suited to microscopy and biomedical tasks where multiple labels can be biologically plausible (Wieslander et al., 2020; Lu et al., 2022; Wundram et al., 2024). However, most CP pipelines assume a flat label space and rely solely on probability-based nonconformity, overlooking (i) biomedical taxonomies (coarse→fine) and (ii) morphology cues that carry discriminative signal in microscopy (Hengst et al., 2025; Olsson et al., 2022).

To address these gaps, we introduce *MR–HCP*, a morphology-regularized *hierarchical* conformal predictor tailored to fine-grained TEM analysis. MR–HCP first performs a super-category conformal gate aligned with the biomedical taxonomy and then refines within retained groups using a morphology-aware nonconformity score that penalizes deviations from class-specific prototypes. We integrate MR–HCP into an end-to-end pipeline with a YOLOv11 detector to demonstrate practicality for single-cell neutrophil ultrastructure TEM (NUS-TEM), including semi-automatic annotation where singleton sets can be auto-accepted and multi-label sets triaged to experts. GT-crop

experiments form the core label-level evaluation of MR–HCP, while YOLO-based results serve as an application study of detector-induced distribution shift.

On a curated dataset ($\sim$4.9k annotations), MR–HCP attains near-nominal coverage with compact sets and high singleton accuracy, outperforming Split CP, Mondrian CP, and concurrent HCC models(Papadopoulos et al., 2002; Romano et al., 2020; Angelopoulos & Bates, 2021; Hengst et al., 2025). On the public Raabin–WBC dataset(Kouzehkanan et al., 2022), MR–HCP similarly achieves strong coverage–set-size trade-offs compared to the same CP baselines, providing evidence that the proposed morphology-aware hierarchical conformalization transfers across microscopy-based biomedical imaging tasks. Beyond inference, MR–HCP supports *dataset reclassification*: refining coarse 4-class labels into 7 fine classes with calibrated uncertainty. Our contributions are:

- **Method:** a sequential super→fine, leaf-only *hierarchical* CP with a *morphology-regularized* score, combining hierarchical gating and morphology beyond flat, probability-only CP and mixed-level node-set HCP baselines, that preserves validity while shrinking sets in visually ambiguous cases.

- **System:** an end-to-end detector→MR–HCP pipeline for TEM (with expert-in-the-loop) to enable trustworthy automation in practice.

- **Evidence:** consistent improvements over CP baselines on biomedical microscopy, with analyses of ambiguity patterns that align with biology (e.g., empty vs. emptying vesicles; primary vs. Specific (secondary) granules).

This work builds on our prior microscopy efforts toward trustworthy annotation in electron and digital pathology (Ahmad et al., 2025b;a), extending them with calibrated uncertainty that is aligned with both the domain taxonomy and morphology.

## 2 RELATED WORK

Confidence estimation via Bayesian approximations, heteroscedastic modeling, and deep ensembles improves *relative* calibration but lacks finite-sample guarantees and can be brittle under shift (Gal & Ghahramani, 2016; Kendall & Gal, 2017; Lakshminarayanan et al., 2017; Gawlikowski et al., 2023). Conformal prediction (CP) instead provides distribution-free, finite-sample coverage by outputting calibrated *sets* (Vovk et al., 2005; Angelopoulos & Bates, 2021). In imaging, CP has been adapted to classification, detection, and segmentation, with variants that trade coverage and set size (Abdar et al., 2021; Romano et al., 2020; Tyagi & Guo, 2023; Katsios & Papadopoulos, 2024; Zhou et al., 2025; Wundram et al., 2024; Bereska et al., 2025; Katsios & Papadopoulos, 2025; Alijani & Najjaran, 2025). Canonical baselines include Split CP (Papadopoulos et al., 2002), Mondrian CP for class-conditional calibration (Vovk et al., 2005; Ding et al., 2023), and APS for cumulative-mass selection (Romano et al., 2020).

Hierarchical classification leverages taxonomies to improve interpretability and align decisions with domain ontologies (Hosseini et al., 2019; Goren et al., 2024). In medical imaging, coarse-to-fine models reflect diagnostic workflow and can ease expert validation (Shafique et al., 2024; Hengst et al., 2025; Lambert et al., 2024). Yet most CP pipelines operate on *flat* labels and do not calibrate across taxonomic levels (Hengst et al., 2025), particularly when fine-grained, leaf-level guarantees are required for expert workflows. Our proposed MR–HCP addresses this gap with a two-stage conformalization (super→fine) that preserves guarantees while reducing set size by filtering implausible super-groups early.

Concurrent to our work, (Hengst et al., 2025) propose Hierarchical Conformal Classification (HCC), which constructs prediction sets over tree nodes (internal and leaf) and guarantees coverage in terms of the union of leaves covered by the selected node set. HCC calibrates multiple node-set covers in parallel and chooses a minimum-cost cover under a hierarchical coverage constraint. In contrast, MR–HCP produces *leaf-only* prediction sets via a sequential two-stage procedure (coarse super-category gate followed by fine-level Mondrian refinement), yielding a different uncertainty object and evaluation target. We therefore treat HCC as the closest hierarchical CP baseline and include a direct quantitative comparison in our experiments.

Handcrafted morphology and texture descriptors remain informative in microscopy where shape, size, and intensity patterns are diagnostic; however, CP methods rarely integrate such cues into the score itself (Olsson et al., 2022). Our morphology-regularized nonconformity injects domain signal

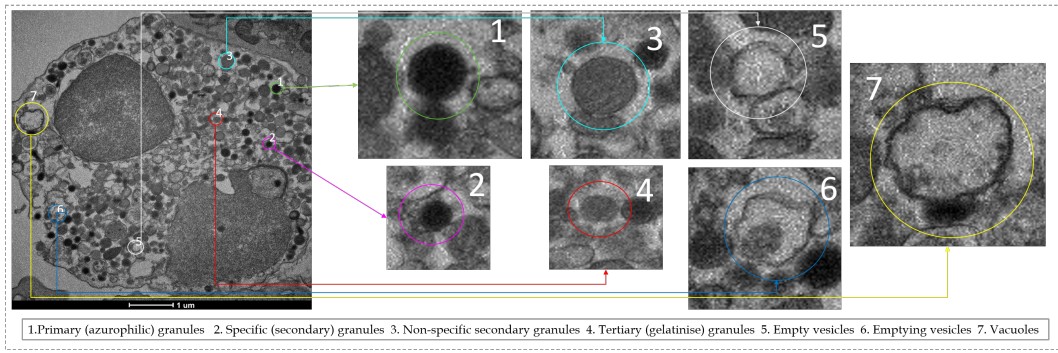

1.Primary (azurophilic) granules  2. Specific (secondary) granules  3. Non-specific secondary granules  4. Tertiary (gelatinise) granules  5. Empty vesicles  6. Emptying vesicles  7. Vacuoles

Figure 1: TEM image of neutrophil ultrastructure highlighting the seven fine classes.

via a clipped Mahalanobis penalty, improving sharpness without sacrificing validity. This complements prior microscopy pipelines and our earlier work on scalable annotation and uncertainty-aware analysis (Ahmad et al., 2025b;a).

Recent works study CP for structured outputs, addressing NMS, localization noise, and pixelwise calibration (Mukama et al., 2025; Timans et al., 2024; Zhou et al., 2025; Wundram et al., 2024; Bereska et al., 2025). We adopt a pragmatic route: pair a YOLO detector with MR–HCP at the label level, calibrating thresholds on *detections* to match test distribution. Our detector coupling is complementary to stronger architectures; MR–HCP itself is agnostic to the underlying detector and can be paired with improved detection backbones in future work.

## 3 METHODOLOGY

We develop a conformal prediction framework that combines hierarchical taxonomies with morphology-aware nonconformity to improve fine-grained classification under uncertainty. While our approach is demonstrated on neutrophil ultrastructure analysis in TEM, the design is broadly applicable to domains where categories are hierarchically organized and morphology- or feature-rich. This section is structured as follows. Section 3.1 introduces notation and the task formulation. Section 3.2 describes the base classifier trained on morphology features. Section 3.3 overviews a preliminary exploratory analysis of morphology features. Section 3.4 presents our proposed *MR-HCP*, which combines hierarchical two-stage conformalization with morphology-aware nonconformity scores. Section 3.5 introduces image-conditional calibration, an orthogonal extension to handle TEM-specific intensity variability. Together, these components constitute a principled and domain-tailored CP framework for neutrophil TEM analysis.

### 3.1 PROBLEM SETUP AND NOTATION

Each instance $x$ corresponds to a candidate ultrastructure segmented from a neutrophil TEM image. We represent $x$ in two complementary ways: (i) the raw intensity patch surrounding the structure, and (ii) derived morphological features $\phi(x) \in \mathbb{R}^d$ computed from its segmentation mask. The prediction task is multiclass classification over $\mathcal{Y} = \{1, \ldots, 7\}$, the seven fine-grained ultrastructure categories (Fig. 1) verified by biomedical experts (Specific (secondary) granules, Primary (azurophilic) granules, Non-specific secondary granules, Tertiary (gelatinise) granules, Empty vesicles, Emptying vesicles, Vacuoles), Ahmad et al. (2025b).

For interpretability and biological alignment, these categories are grouped into four super-categories $\mathcal{G} = \{1, \ldots, 4\}$, namely Light Granules, Dark Granules, Vesicles, and Vacuoles, with $\mathcal{Y}_g \subset \mathcal{Y}$ denoting the fine classes belonging to each super-category. This hierarchical taxonomy is central to our conformal design: we first decide among $\mathcal{G}$ and then refine within the chosen subset $\mathcal{Y}_g$. The overall end-to-end pipeline is summarized in Fig. 2.

### 3.2 BASE PROBABILISTIC CLASSIFIER

We use a feature-based multiclass LightGBM trained on morphology+intensity descriptors extracted from expert masks (geometry, shape, intensity statistics, and GLCM features; details in Appx. E). Training is image-wise split to avoid leakage; class imbalance is handled via inverse-frequency

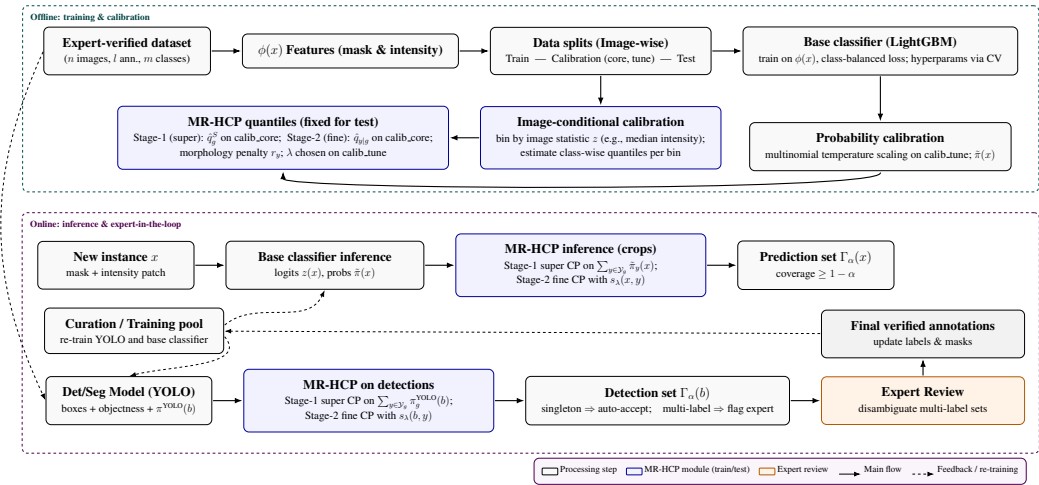

Figure 2: End-to-end pipeline. *Offline (top):* expert-verified images are processed through feature extraction, dataset splitting, and base classifier training. Calibration includes temperature scaling, image-conditional binning, and MR–HCP quantile estimation, fixing thresholds before deployment. *Online (bottom):* YOLO detections or cropped instances are passed through MR–HCP. Singleton outputs are accepted automatically, while multi-label sets are flagged for expert triage in CVAT. Verified annotations are returned to a curation pool, enabling iterative retraining and refinement.

weights. We apply multinomial temperature scaling on a held-out calibration slice to stabilize probabilities used by all CP variants (including MR–HCP). Hyperparameters, preprocessing, and full equations for probability calibration are reported in Appx. A.

### 3.3 EXPLORATORY ANALYSIS OF MORPHOLOGY–INTENSITY FEATURES

We extract 19 handcrafted descriptors from expert-verified masks spanning geometry/shape, intensity statistics, and GLCM texture (feature list and computation details in Appx. E). On our 4,899 annotated instances (7 fine classes), a 2D PCA projection (62.5% variance) shows partial grouping—e.g., Vacuoles are well separated and Tertiary (gelatinise) granules are compact—while Primary (azurophilic) and Specific (secondary) granules overlap substantially; a UMAP view further reveals entanglement between vesicles and granules (Appx. Fig. 7). These trends indicate that hand-crafted features carry discriminative signal yet cannot fully resolve certain fine subtypes, motivating our hierarchy-plus-morphology design in MR–HCP. A correlation analysis of the 19 descriptors (Appx. C, Table 5) shows only modest pairwise correlations between morphology, intensity, and texture blocks, supporting their use as complementary inputs for both the base classifier and the morphology penalty.

### 3.4 MORPHOLOGY-REGULARIZED HIERARCHICAL CONFORMAL PREDICTION (MR-HCP)

Hierarchical Conformal Prediction (HCP) adapts the split conformal framework to problems with taxonomic label structures. In its standard form, prediction proceeds in two stages: (i) conformalization at the level of coarse categories, followed by (ii) refinement within the retained subsets of fine classes. This hierarchical decomposition helps align statistical guarantees with domain ontologies and often yields smaller, more interpretable prediction sets than flat conformalization. The detailed two-stage MR–HCP procedure is illustrated in Fig. 3

**Stage 1: Super-category conformalization.** Let $\mathcal{G} = \{1, \ldots, 4\}$ denote the super-categories (Light Granules, Dark Granules, Vesicles, Vacuoles), each associated with a disjoint set of fine classes $\mathcal{Y}_g \subset \mathcal{Y}$. We aggregate softmax probabilities (Appx equation 13) across super-categories:

$$\pi_g^S(x) = \sum_{y \in \mathcal{Y}_g} \pi_y(x), \qquad g \in \mathcal{G}. \tag{1}$$

The nonconformity score at this level is

$$s_{\sup}(x, g) = 1 - \pi_g^S(x), \tag{2}$$

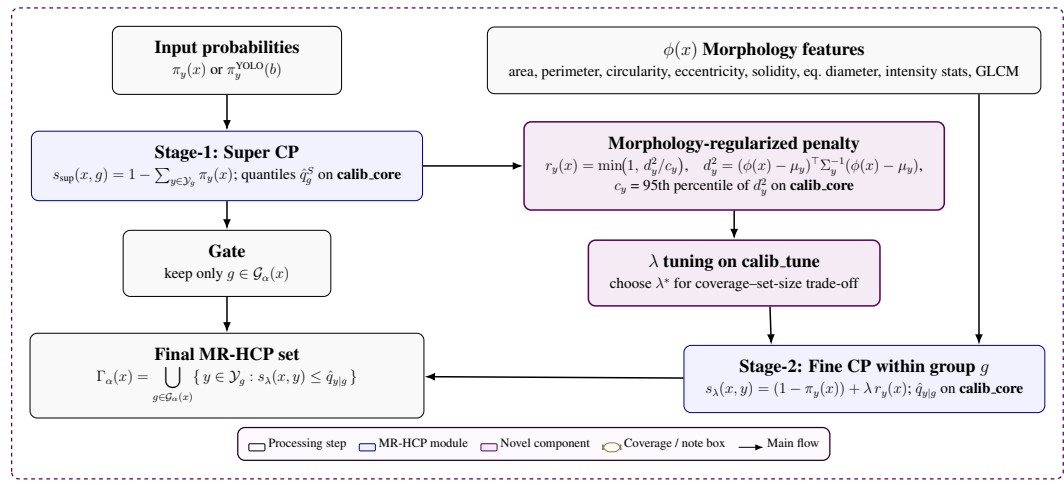

Figure 3: *MR-HCP*. Stage-1 aggregates probabilities into super-categories and applies conformal thresholds to filter plausible groups. Stage-2 refines within retained groups using morphology-regularized nonconformity scores that penalize deviations from class prototypes. Two novel components are highlighted: (i) morphology penalty based on Mahalanobis distance, and (ii) $\lambda$ tuning to balance coverage and set size. Final prediction sets combine both stages to yield compact outputs with formal coverage guarantees.

and the retained super-categories are those with score below the calibrated quantile $\hat{q}_g^S$:

$$\mathcal{G}_\alpha(x) \; = \; \{g \in \mathcal{G} : s_{\sup}(x,g) \leq \hat{q}_g^S\}. \tag{3}$$

**Stage 2: Fine-class refinement with morphology regularization** Within each retained super-category $g \in \mathcal{G}_\alpha(x)$, standard HCP would refine using the class probabilities $\pi_y(x)$ alone. Instead, we incorporate quantitative morphology features $\phi(x) \in \mathbb{R}^d$ derived from segmentation masks and intensities (area, perimeter, circularity, eccentricity, solidity, equivalent diameter, intensity quartiles/mean/skew, GLCM contrast/homogeneity).

For each fine class $y$, we compute the class-conditional mean $\mu_y$ and covariance $\Sigma_y$ on calibration features. The Mahalanobis distance

$$d_y^2(x) = \big(\phi(x) - \mu_y\big)^\top \Sigma_y^{-1} \big(\phi(x) - \mu_y\big) \tag{4}$$

quantifies morphological deviation. To normalize scales across classes, let $c_y$ denote the 95th percentile of $d_y^2$ for class $y$. We then define the bounded morphology penalty

$$r_y(x) \; = \; \min\Big(1, \, \frac{d_y^2(x)}{c_y}\Big). \tag{5}$$

The morphology-regularized nonconformity score is

$$s_\lambda(x,y) \; = \; \big(1 - \pi_y(x)\big) + \lambda \, r_y(x), \qquad \lambda \geq 0, \tag{6}$$

We adopt this additive form for simplicity and stability; in our ablations (Sec. 4.4), multiplicative and soft-clip variants exhibit very similar behaviour, with the additive score offering a slightly better size–singleton trade-off. where $\lambda$ balances the contribution of morphology relative to classifier confidence. Thresholds $\hat{q}_{y|g}$ are estimated on the calibration set restricted to $\mathcal{Y}_g$, with $\lambda$ fixed *a priori* from a held-out tuning slice.

The final MR-HCP set is

$$\Gamma_\alpha^{\text{MR-HCP}}(x) \; = \; \bigcup_{g \in \mathcal{G}_\alpha(x)} \Big\{ y \in \mathcal{Y}_g : s_\lambda(x,y) \leq \hat{q}_{y|g} \Big\}. \tag{7}$$

We denote by $\alpha_{\sup}$ and $\alpha_{\text{sub}}$ the miscoverage budgets allocated to the super and fine-level stages, respectively, and explicitly split the overall target as $\alpha = \alpha_{\sup} + \alpha_{\text{sub}}$. We then calibrate the two

stages sequentially: the super-category gate is calibrated at level $\alpha_{\text{sup}}$, and, conditional on passing this gate, the fine-level predictors within each retained super-category are calibrated at level $\alpha_{\text{sub}}$. By the union-bound argument, MR–HCP maintains finite-sample coverage:

$$\mathbb{P}\big(Y \in \Gamma_\alpha^{\text{MR-HCP}}(X)\big) \ \geq \ 1 - \alpha_{\text{sup}} - \alpha_{\text{sub}}, \tag{8}$$

under exchangeability and with $\lambda$ fixed prior to quantile estimation. Thus, MR-HCP inherits the validity of split CP while leveraging morphology to refine uncertainty sets.

Unlike prior HCP approaches, which rely solely on probabilistic scores from a base classifier, MR-HCP integrates domain-specific morphology descriptors into the nonconformity function. This modification has two benefits: (i) *biological plausibility*, since misclassified ultrastructures often differ subtly in size or shape despite similar intensity profiles; and (ii) *statistical sharpness*, as morphology regularization reduces large but spurious prediction sets in visually ambiguous cases. To our knowledge, this is the first conformal prediction framework explicitly combining hierarchical taxonomy with morphology-aware nonconformity in biomedical electron microscopy.

### 3.5 IMAGE-CONDITIONAL CALIBRATION

TEM images often exhibit intensity shifts across sessions or staining conditions, which can degrade coverage if ignored. To address this, we integrate an *image-conditional calibration* strategy, which stratifies calibration data into bins $\{B_k\}$ according to a global statistic $z(x)$ (e.g., image median intensity). Within each bin, class-wise quantiles $\hat{q}_{y|B_k}$ are estimated. At test time, a sample $x$ is assigned to its closest bin, and the corresponding thresholds are applied:

$$\hat{q}_y(x) \ = \ \hat{q}_{y|B_k}, \quad x \in B_k. \tag{9}$$

This Mondrian-style partition yields approximately conditional coverage within bins under exchangeability.

We treat image-conditional calibration as an *orthogonal extension* to MR-HCP: it does not alter the two-stage hierarchical structure, but rather improves robustness to dataset-specific imaging variability. In practice, we find that combining MR-HCP with image-conditional calibration achieves the most stable coverage across the diverse intensity regimes present in neutrophil TEM.

### 3.6 INTEGRATION WITH YOLO DETECTION (COARSE→FINE)

We integrate MR–HCP with a YOLOv11(-seg) detector to obtain an end-to-end pipeline that localizes instances and returns *calibrated* fine-level label sets (Fig. 4). At inference, each detection $b$ provides a mask/box and coarse logits; we also extract morphology features $\phi(b)$ and calibrated fine probabilities $\tilde{\pi}(b)$ from the feature classifier.

**Detector–classifier fusion at the super gate.** To decide which super-groups are admissible before fine refinement, we fuse detector coarse beliefs with aggregated fine beliefs:

$$\bar{\pi}_g^S(b) = \eta\, \pi_g^{\text{YOLO}}(b) + (1-\eta) \sum_{y \in \mathcal{Y}_g} \tilde{\pi}_y(b), \quad s_{\text{sup}}(b,g) = 1 - \bar{\pi}_g^S(b), \tag{10}$$

where $\eta \in [0,1]$ is fixed on a disjoint tuning slice. Split-CP thresholds $\{\hat{q}_g^S\}$ are then estimated on a *calibration set of matched detections* (not on GT crops) to mirror test-time conditions.

**Morphology-regularized fine score.** Within each retained super-group, MR–HCP refines using the morphology-aware score

$$s_\lambda(b,y) = \big(1 - \tilde{\pi}_y(b)\big) + \lambda\, r_y(b), \qquad r_y(b) = \min\big(1,\, d_y^2(b)/c_y\big), \tag{11}$$

where $d_y^2$ is the Mahalanobis distance of $\phi(b)$ from the class prototype and $c_y$ (95th percentile) normalizes scale; $\lambda$ is tuned on a calibration-tuning slice and fixed before estimating fine-level quantiles $\{\hat{q}_{y|g}\}$.

The result is a two-stage, detection-aware MR–HCP that (i) filters implausible super-groups with fused evidence and (ii) sharpens fine sets via morphology. All remaining implementation details (Hungarian matching and IoU criterion, calibration protocol, image-conditional binning, ablations on $\eta/\lambda$ and operating-point sweeps) are provided in the Appendix (see Appx. B and Appx. F).

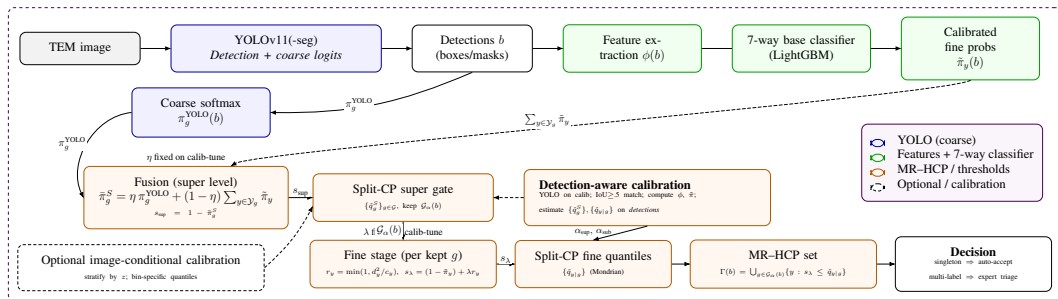

Figure 4: Compact layout of the YOLO→MR–HCP pipeline. The detections fork is kept tight under *Detections*; fusion sits centered beneath it; the main MR–HCP stages form a 2×2 grid, with calibration options below. Singletons are auto-accepted; multi-label sets are triaged to experts.

**Detector training (coarse labels).** Let $\mathcal{G} = \{$Light Granules, Dark Granules, Vesicles, Vacuoles$\}$ denote the super-categories (We use 4 super-classes by default; in low-support regimes one may drop Vacuoles and handle them at Stage-2 only). YOLOv11(-seg) is trained *image-wise* on $\mathcal{G}$ to maximize localization recall and segmentation quality. At inference, each detection $b$ (box/mask) produces coarse logits $z^{\text{YOLO}}(b) \in \mathbb{R}^{|\mathcal{G}|}$ and softmax probabilities

$$\pi_g^{\text{YOLO}}(b) \;=\; \frac{\exp\big(z_g^{\text{YOLO}}(b)\big)}{\sum_{h \in \mathcal{G}} \exp\big(z_h^{\text{YOLO}}(b)\big)} \;, \qquad g \in \mathcal{G}. \tag{12}$$

## 4 RESULTS

### 4.1 EXPERIMENTAL SETUP

Unless noted otherwise, all numbers are reported on the held-out *TEST* split. Following Section 3.4, we evaluate (i) standard Split/Mondrian CP, (ii) APS (global and bucketed), (iii) Morphology CP (flat, no hierarchy), (iv) HCC, and (v) our proposed MR–HCP. The base probabilistic model is a feature-based LightGBM trained on morphology+intensity descriptors (Section 3.2), with multinomial temperature scaling applied on the calibration tuning set. For the hierarchical method, we use the super/fine taxonomy in Section 3.1 and morphology-regularized scores in Stage 2 ( equation 6).

We sweep $(\alpha_{\text{sup}}, \alpha_{\text{sub}})$ and select the point marked in Fig. 6: $(\alpha_{\text{sup}}, \alpha_{\text{sub}}) = (0.10, 0.05)$, which attains near-nominal coverage with compact sets on NUS–TEM. Unless stated, MR–HCP results refer to this operating point, and the reported numbers refer to the NUS–TEM dataset.

### 4.2 GLOBAL COMPARISON WITH BASELINES

Table 1 summarizes coverage and set efficiency at a nominal target of 0.90 across both NUS–TEM and Raabin–WBC. On NUS–TEM, MR–HCP achieves a favourable coverage–efficiency balance: achieved coverage 0.937 with average set size 1.12, yielding a high singleton fraction (0.888) at 0.932 singleton accuracy (the full coverage–size frontier across nominal targets on NUS–TEM, including all baselines, is shown in Appx. C, Fig. 8). APS variants meet or exceed the target coverage but inflate sets (on the order of 5–6 labels on average), while Mondrian CP under-covers (0.919) despite modest sets; HCC attains very high coverage (0.994) but with substantially larger prediction sets (4.48 on average) and far fewer singletons. On Raabin–WBC, MR–HCP again attains strong coverage–set-size tradeoffs (coverage 0.944, average set size 1.39), outperforming Mondrian CP and APS at the same target level, while HCC achieves slightly higher coverage at the cost of larger sets and a lower singleton fraction than MR–HCP. Together, these results indicate that the morphology-regularized hierarchical design provides compact, leaf-level prediction sets on two microscopy datasets while remaining competitive with both flat CP baselines and concurrent hierarchical HCC. Bootstrap confidence intervals with 1000 resamples on NUS–TEM (GT crops) confirm that these MR–HCP metrics (coverage, average set size, singleton accuracy) are stable; see Appx. C.

### 4.3 STAGE-WISE MR–HCP PERFORMANCE

Using only super-category conformalization (Stage 1, no fine refinement), MR–HCP attains achieved coverage 0.962 with average set size 1.039 (singleton fraction 0.965; singleton accuracy 0.961), confirming that coarse grouping captures most discriminative signal while yielding

Table 1: Global comparison at nominal target 0.90 on NUS–TEM and Raabin–WBC datasets. MR–HCP achieves favourable coverage–efficiency tradeoffs on both datasets compared to CP baselines and concurrent hierarchical HCC.

| | NUS–TEM | | | | Raabin–WBC | | | |
|---|---|---|---|---|---|---|---|---|
| Method | Achieved cov. | Avg set size | Singleton frac. | Singleton acc. | Achieved cov. | Avg set size | Singleton frac. | Singleton acc. |
| Mondrian CP | 0.919 | 1.85 | 0.438 | 0.892 | 0.891 | 1.53 | 0.624 | 0.902 |
| APS (global) | 0.978 | 5.85 | 0.065 | 0.839 | 0.993 | 4.12 | 0.050 | 0.972 |
| APS (bucketed) | 0.968 | 5.27 | 0.139 | 0.856 | 0.990 | 4.11 | 0.048 | 0.971 |
| Morphology CP | 0.836 | 1.522 | 0.601 | 0.809 | 0.891 | 1.535 | 0.624 | 0.902 |
| HCC (Hengst et al., 2025) | 0.994 | 4.48 | 0.055 | 0.981 | 0.998 | 3.09 | 0.053 | 0.991 |
| **MR–HCP (ours)** | **0.937** | **1.12** | **0.888** | **0.932** | **0.944** | **1.39** | **0.725** | **0.946** |

Table 2: Coverage and average set size at the super-category level. We report both (i) Stage 1 conformalization directly on the super-classes, and (ii) aggregated results at the same super-class level after full MR–HCP (two-stage) refinement. This comparison highlights how Stage 2 refinement slightly alters coverage and set size distributions while preserving validity.

| Super Class | Stage 1 (super only) | | Stage 2 (aggregated) | |
|---|---|---|---|---|
| | Coverage | Avg set size | Coverage | Avg set size |
| Light Granules | 0.9841 | 1.068 | 0.9483 | 1.078 |
| Dark Granules | 0.8974 | 1.009 | 0.8718 | 1.013 |
| Vesicles | 0.9836 | 1.000 | 0.9836 | 1.399 |
| Vacuoles | 1.0000 | 1.000 | 1.0000 | 1.000 |

highly compact sets (Table 2). After Stage 2 refinement at the tuned operating point $(\alpha_{\text{sup}}, \alpha_{\text{sub}}) = (0.10, 0.05)$, the method maintains near-nominal coverage with compact sets (Table 1), while enabling fine-grained, leaf-level prediction. The most notable shift at the super level occurs for Vesicles (average set size rises from $1.000$ in Stage 1 to $1.399$ after aggregation) due to systematic ambiguity between Empty vs. Emptying subtypes, while at the fine level *Emptying vesicles* require the largest sets ($\approx 1.600$ on average) and *Vacuoles* are consistently singletons (coverage $\approx 1.000$), reflecting distinctive morphology. Detailed per-super and per-class results and additional sensitivity analyses appear in Appx. C.

### 4.4 ABLATIONS ON SCORE DESIGN AND HIERARCHY

We ablate the contribution of morphology and hierarchy, as well as alternative score combinations (multiplicative, soft-clip) for the Stage 2 score. As summarized in Table 7 (Appx. C), hierarchy-only ($\lambda=0$), multiplicative, and soft-clip variants yield comparable coverage, but the full MR–HCP additive score attains the smallest average set size and highest singleton fraction, confirming that morphology provides complementary information beyond hierarchy alone.

### 4.5 OPERATING-POINT TUNING: COVERAGE VS TARGET

Fig. 6 plots achieved coverage against the nominal target for different $(\alpha_{\text{sup}}, \alpha_{\text{sub}})$ pairs. The selected operating point $(\alpha_{\text{sup}}, \alpha_{\text{sub}}) = (0.10, 0.05)$ achieves near-nominal coverage (0.937) with compact sets (1.12 average size), striking the best coverage–efficiency tradeoff among the configurations we considered. Alternative points show predictable trends: lower $\alpha$ values increase coverage but enlarge sets; higher $\alpha$ values reduce coverage and shrink sets. Details of the small grid used for selection are reported in Appx. C.

### 4.6 YOLO→MR–HCP UNDER DETECTION MATCHING

We evaluate the integrated detector–calibrator pipeline on detections matched to ground truth at IoU$\geq 0.5$ (TEST matched fraction $\approx 0.738$). At our selected operating point, the end-to-end system attains competitive coverage with substantially reduced average set size relative to detector-only confidence heuristics, while the underlying YOLOv11(-seg) baseline provides solid localization quality (box mAP50$= 0.533$, mask mAP50$= 0.533$) with strongest performance on Vesicles and Dark Granules and weaker scores for rare Vacuoles. Matched-only summaries differ from raw-inference aggregates because coverage is computed *conditioned on successful localization*, which is the correct basis for label-set validity. Ablations show that adding morphology regularization ($\lambda > 0$) reduces set size with small coverage changes, and optional image-conditional binning yields modest

Table 3: Fine classes (Stage 2). Coverage and set size per class at the tuned point $(0.10, 0.05)$.

| Fine class | Coverage | Avg set size |
|---|---|---|
| Tertiary (gelatinise) granules | 0.9573 | 1.050 |
| Non-specific secondary granules | 0.9369 | 1.113 |
| Specific (secondary) granules | 0.8537 | 1.008 |
| Primary (azurophilic) granules | 0.8919 | 1.018 |
| Empty vesicles | 0.9898 | 1.224 |
| Emptying vesicles | 0.9765 | 1.600 |
| Vacuoles | 1.0000 | 1.000 |

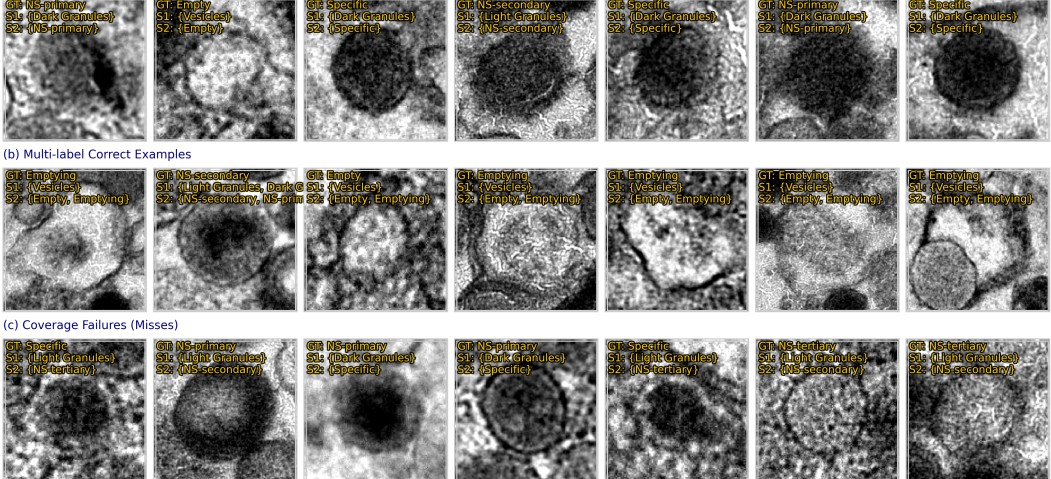

Figure 5: Qualitative MR–HCP outcomes on neutrophil TEM images. Each panel compares ground-truth labels (GT) with MR–HCP outputs at the super-class level (S1) and fine-class level (S2). (*a*) Singleton-correct cases where the conformal set reduces to the true label; (*b*) Multi-label sets that capture biologically plausible alternatives; (*c*) Rare misses (failure) where the true label is absent.

robustness gains at near-constant set size. Full detection-coupled tables, per-class breakdowns, and sweep/ablation figures are provided in Appx. D.

To isolate the impact of detection errors, we decompose coverage into matched and unmatched YOLO detections (Table 11, Appx. D). On matched detections (96% of test instances), MR–HCP attains high coverage (0.966) with compact sets (average $|\Gamma|=1.12$), whereas on unmatched cases coverage drops to 0.189 at similar average set size (average $|\Gamma|=1.09$). This decomposition indicates that the dominant coverage loss in the end-to-end pipeline stems from detector-induced distribution shift rather than from the conformalization mechanism itself.

## 4.7 QUALITATIVE OUTCOMES

Fig. 5 illustrates representative qualitative outcomes that complement the quantitative results. The majority of detections resolve to confident singletons ($\sim$89%, Table 1), such as Primary (azurophilic), non-specific secondary, or empty vesicles (Fig. 5a), highlighting the reliability of MR–HCP when predictions are unambiguous. In cases of biological ambiguity, the conformal sets expand modestly to include multiple plausible alternatives (Fig. 5b). These multi-label sets provide calibrated transparency rather than overconfidence, ensuring that the true class remains included in nearly all cases. Coverage misses remain relatively rare ($\approx 6\%$ at the chosen operating point) and typically arise in highly entangled subtypes, underscoring the intrinsic difficulty of certain fine-grained distinctions even for experts. Together, these outcomes show that MR–HCP produces compact and interpretable sets that either collapse to a confident singleton or, in ambiguous regions, transparently communicate uncertainty in a biologically meaningful way.

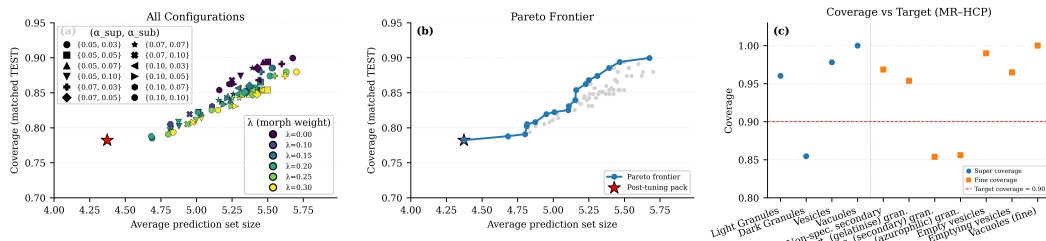

Figure 6: (*a*) YOLO→MR–HCP sweep on matched detections. Coverage vs. average set size for all sweep configurations; marker encodes $(\alpha_{\text{sup}}, \alpha_{\text{sub}})$ and color encodes $\lambda$. The star marks the post-tuning pack used at inference, $(\alpha_{sup}, \alpha_{sub}) = (0.07, 0.07)$ for detector-coupled results. *(b)* Pareto frontier (maximize coverage, minimize set size) with the chosen pack highlighted. (*c*) Coverage vs. target for MR–HCP, sweeping over super vs. fine levels with the selected operating point $(0.10, 0.05)$ used throughout the main NUS–TEM results.

## 5 DISCUSSION

**Why hierarchy and morphology help.** The super-gate removes implausible families early, so fine calibration compares only biologically coherent candidates. The morphology term $r_y$ penalizes shape/size/intensity profiles that are atypical for a class, shrinking spurious alternatives when probabilities alone are indecisive and preserving singletons when both cues agree. Empirically, this yields compact, interpretable sets at near-nominal coverage (Table 1; Appx. results).

**Ambiguity mirrors biology.** Largest sets arise for *Emptying vesicles* and dark-granule sub-classes—exactly where morphology and intensity overlap (empty vs. emptying; specific vs. primary). MR–HCP exposes this ambiguity rather than collapsing to brittle single labels, which is preferable for expert review and scientific integrity.

**Validity under detector coupling.** We estimate thresholds on *detections* matched to ground truth, aligning calibration and test distributions. Coverage statements are thus conditioned on successful localization (IoU$\geq$0.5). Differences between matched-only and raw inference summaries follow from this conditioning and are reported transparently in the appendix.

**Limitations.** (1) The morphology penalty uses handcrafted descriptors and Mahalanobis statistics; deep embeddings could capture subtler cues. (2) The hierarchy is hard-gated: super-gate errors cannot be recovered downstream. (3) Guarantees are conditioned on the detector; missed localizations are out of scope for CP at the label level.

**Future work.** (i) Replace handcrafted features with self-supervised or foundation embeddings (Oquab et al.), (ii) explore soft or recoverable hierarchies, (iii) incorporate detector uncertainty into calibration, and (iv) validate across larger multi-institutional TEM cohorts and other immune cell types. Finally, we will leverage MR–HCP to accelerate *reclassification* of partially refined datasets by auto-accepting singleton sets and triaging the rest.

## 6 CONCLUSION

We presented *MR–HCP*, a morphology-regularized *hierarchical* conformal predictor for fine-grained TEM ultrastructure analysis. By combining a super-category conformal gate with a morphology-aware fine score, MR–HCP delivers compact, calibrated prediction sets that align with biological structure. Integrated with YOLO, the system supports practical expert-in-the-loop workflows—auto-accepting singletons and flagging ambiguity—while outperforming standard CP baselines in coverage, set size, and singleton accuracy. Beyond inference, MR–HCP enables reliable *dataset reclassification* from coarse to fine labels with quantified uncertainty, offering a scalable path for building richer microscopy datasets.

### ACKNOWLEDGMENT

This work made limited use of large language model tools for early code prototyping and minor editorial suggestions. All ideas, methodology, experiments, analyses, and final text are the authors' own; model outputs were reviewed and edited by the authors.

REPRODUCIBILITY

We provide a complete description of the task, label taxonomy, and notation in Sections 3.1–3.4, with the image-conditional calibration variant in Section 3.5 and the detector coupling in Section 3.6. Implementation details for the base classifier, probability calibration, and feature computation are in Appx. A and E; algorithmic pseudocode for calibration/inference (both crops-only and YOLO→MR–HCP) appears in Appx. F. The detection-aware calibration protocol (Hungarian matching, IoU thresholds, fused super-gate) and additional analyses are documented in Appx. B and D. Operating-point sweeps, image-conditional binning, and morphology-weight ablations are reported with figures/tables in Appx. C; split definitions and counts are in Appx. E. We use image-wise splits to avoid leakage, and apply standard randomized tie-breaking at quantile thresholds (Appx. E). Upon acceptance, we will release code, configuration files, and scripts to reproduce all tables/figures, along with instructions to regenerate features and prediction sets from the raw data (or to re-compute them where redistribution is restricted).

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

## APPENDIX

## A  BASE PROBABILISTIC CLASSIFIER: DETAILS

We employ a gradient-boosted decision tree model (LightGBM) trained on the morphology feature set $\phi(x)$. This choice is motivated by (i) its ability to capture nonlinear interactions among geometric and intensity-based descriptors, and (ii) its efficiency on medium-sized biomedical datasets. The model outputs logits $z(x) \in \mathbb{R}^K$ which are converted to predictive probabilities via the softmax:

$$\pi_y(x) = \frac{\exp(z_y(x))}{\sum_{j=1}^{K} \exp(z_j(x))}, \qquad y \in \{1, \ldots, K\}. \tag{13}$$

We train with multiclass cross-entropy loss and class weights inversely proportional to class frequencies to mitigate imbalance. Hyperparameters (e.g., *num_leaves*, *max_depth*, *learning_rate*) are

selected by stratified image-wise cross-validation and then fixed for the final model trained on the full training set.

To avoid information leakage, all splits are performed at the image level: *train*, *calibration*, and *test*. The calibration split is further divided into a core subset (for quantile estimation) and a tuning subset (for hyperparameters such as $\lambda$ in morphology-regularization and bin edges in image-conditional calibration).

Although conformal prediction does not assume calibrated probabilities, we improve stability by applying multinomial temperature scaling on the calibration tuning set. The adjusted probabilities are

$$\tilde{\pi}_y(x; T) \;=\; \frac{\exp\big(z_y(x)/T\big)}{\sum_{j=1}^{K} \exp\big(z_j(x)/T\big)}, \tag{14}$$

where $T^\star$ minimizes multiclass cross-entropy. The calibrated $\tilde{\pi}(x)$ serve as inputs for all conformal variants, including our proposed MR-HCP.

## B  INTEGRATION WITH YOLO DETECTION: DETAILS

**Detector training (coarse labels).**  Let $\mathcal{G} = \{\text{Light Granules, Dark Granules, Vesicles, Vacuoles}\}$ denote the super-categories (we use 4 super-classes by default; in low-support regimes one may drop Vacuoles and handle them at Stage-2 only). YOLOv11(-seg) is trained *image-wise* on $\mathcal{G}$ to maximize localization recall and segmentation quality. At inference, each detection $b$ (box/mask) produces coarse logits $z^{\text{YOLO}}(b) \in \mathbb{R}^{|\mathcal{G}|}$ and softmax probabilities

$$\pi_g^{\text{YOLO}}(b) \;=\; \frac{\exp\big(z_g^{\text{YOLO}}(b)\big)}{\sum_{h \in \mathcal{G}} \exp\big(z_h^{\text{YOLO}}(b)\big)}, \qquad g \in \mathcal{G}. \tag{15}$$

**Fine classifier on morphology+intensity features.**  Independently, we train a 7-way *feature-based* classifier (LightGBM) on expert-verified masks for the fine classes $\mathcal{Y} = \{1, \ldots, 7\}$. Each instance is represented by a feature vector $\phi(x) \in \mathbb{R}^d$ encoding morphology and intensity descriptors: area, perimeter, equivalent diameter, circularity, solidity, eccentricity, intensity quartiles/mean/IQR/skew, and GLCM contrast/homogeneity. At test time, for each detection $b$ we extract $\phi(b)$ from the YOLO mask and obtain calibrated fine-class probabilities $\tilde{\pi}_y(b)$ (using temperature scaling on a calibration slice). The super-class partition $\{\mathcal{Y}_g\}_{g \in \mathcal{G}}$ is fixed (Light: secondary/tertiary; Dark: specific/primary; Vesicles: empty/emptying; Vacuoles: vacuoles).

**Super-gate with detector–classifier fusion (novel).**  To decide which super-groups are admissible before fine refinement, we fuse the detector's coarse belief with the classifier's aggregated fine belief via a convex combination

$$\bar{\pi}_g^S(b) \;=\; \eta\, \pi_g^{\text{YOLO}}(b) \;+\; (1 - \eta) \sum_{y \in \mathcal{Y}_g} \tilde{\pi}_y(b), \qquad \eta \in [0, 1], \tag{16}$$

and define the super-level nonconformity

$$s_{\sup}(b, g) \;=\; 1 - \bar{\pi}_g^S(b). \tag{17}$$

The weight $\eta$ is selected on a disjoint calibration–tuning split to trade off coverage vs. set size; once fixed, split-CP validity is preserved. On a held-out *calibration-core* set of detections with true super-labels, we compute the $(1 - \alpha_{\sup})$ class-conditional quantiles $\{\hat{q}_g^S\}$ and retain

$$\mathcal{G}_\alpha(b) \;=\; \big\{\, g \in \mathcal{G} \,:\, s_{\sup}(b, g) \leq \hat{q}_g^S \,\big\}. \tag{18}$$

**Fine-stage morphology–regularized score.**  Within each retained super-group $g \in \mathcal{G}_\alpha(b)$, we apply the morphology-regularized score

$$d_y^2(b) \;=\; \big(\phi(b) - \mu_y\big)^\top \Sigma_y^{-1} \big(\phi(b) - \mu_y\big), \tag{19}$$

$$r_y(b) \;=\; \min\Big(1,\, d_y^2(b)/c_y\Big), \tag{20}$$

$$s_\lambda(b, y) \;=\; \big(1 - \tilde{\pi}_y(b)\big) \;+\; \lambda\, r_y(b), \qquad y \in \mathcal{Y}_g, \tag{21}$$

where $(\mu_y, \Sigma_y)$ are class-conditional mean and shrinkage covariance estimated on calibration features, $c_y$ is the 95th percentile of $d_y^2$ per class (for scale normalization), and $\lambda \geq 0$ is tuned on the calibration–tuning split. On *calibration-core* detections with true fine labels, we compute $(1 - \alpha_{\text{sub}})$ Mondrian quantiles $\{\hat{q}_{y|g}\}$ and form the MR–HCP set

$$\Gamma_\alpha^{\text{MR–HCP}}(b) \ = \ \bigcup_{g \in \mathcal{G}_\alpha(b)} \Big\{ y \in \mathcal{Y}_g \ : \ s_\lambda(b, y) \leq \hat{q}_{y|g} \Big\}. \tag{22}$$

Under exchangeability and with $(\eta, \lambda)$ fixed *prior* to quantile estimation, split-conformal validity yields

$$\Pr\big(Y \in \Gamma_\alpha^{\text{MR–HCP}}(B)\big) \ \geq \ 1 - \alpha_{\text{sup}} - \alpha_{\text{sub}}. \tag{23}$$

**Detection-aware calibration protocol.** To match the test-time distribution, all thresholds in equation 18–equation 22 are estimated on *detections*, not on ground-truth crops: (i) run YOLO on calibration images, (ii) associate detections to ground-truth instances via Hungarian matching with IoU $\geq 0.5$, (iii) compute $\phi(b)$ from the YOLO mask, $\tilde{\pi}(b)$ from the fine classifier, and record true labels from the matched ground truth. This produces calibration tuples $\{(b, \phi(b), \tilde{\pi}(b), y)\}$ whose distribution mirrors inference.

**Image-conditional calibration (optional).** To address image-level intensity shifts, we stratify calibration detections by an image statistic $z$ (e.g., median normalized intensity) into bins $\{B_k\}$ and estimate class-wise quantiles $\hat{q}_{g|B_k}^S$ and $\hat{q}_{y|g,B_k}$. At test time, a detection is routed to its nearest bin $B_k$ and the corresponding thresholds are used, yielding approximately conditional coverage within strata.

**Decision logic and expert triage.** If $\Gamma_\alpha^{\text{MR–HCP}}(b)$ is a singleton, we auto-accept the label; otherwise we flag the detection for CVAT review. This concentrates expert time on ambiguous cases while retaining rigorous coverage guarantees for all detections. The expert-verified annotations are fed back to the curation pool to periodically retrain YOLO and the fine classifier.

**Remarks.** (i) If Vacuoles are rare, YOLO may be trained on 4 classes and Vacuoles handled entirely by Stage 2; if Vacuoles are frequent, adding a 5th coarse class can improve the super gate. (ii) If masks are not produced by YOLO, $\phi(b)$ can be computed from a segmentation head or from a high-quality external segmenter; without masks, $r_y(b)$ can be omitted by setting $\lambda = 0$ (ablated baseline). (iii) The fusion in equation 16 is a lightweight novelty that exploits complementary strengths of detector logits and feature-based fine classifier probabilities while preserving split-CP validity (since $\eta$ is fixed on a disjoint slice).

# C   MR–HCP (NO DETECTOR): ADDITIONAL RESULTS

This section complements the main text with operating-point selection, image-conditional calibration, and morphology ablations for MR–HCP without detector coupling. Unless stated otherwise, all numbers are on the held-out *TEST* split. We use the super/fine taxonomy and morphology-regularized Stage 2 score described in the main paper.

## BOOTSTRAP CONFIDENCE INTERVALS

To assess the stability of our main MR–HCP metrics, we compute nonparametric bootstrap confidence intervals on NUS–TEM (GT crops). We resample the TEST set at the instance level 1000 times with replacement and recompute coverage, average set size, and singleton accuracy for each resample. The empirical 2.5th and 97.5th percentiles yield the 95% confidence intervals in Table 4.

Table 4: Bootstrap confidence intervals (1000 iterations) for MR–HCP on NUS–TEM (GT crops).

| Metric | Estimate | 95% CI |
|---|---|---|
| Coverage | 0.9370 | [0.9216, 0.9506] |
| Avg set size | 1.1240 | [1.1031, 1.1440] |
| Singleton acc | 0.9355 | [0.9182, 0.9504] |

The narrow intervals indicate that the reported MR–HCP metrics are stable with respect to sampling variability on the held-out TEST split.

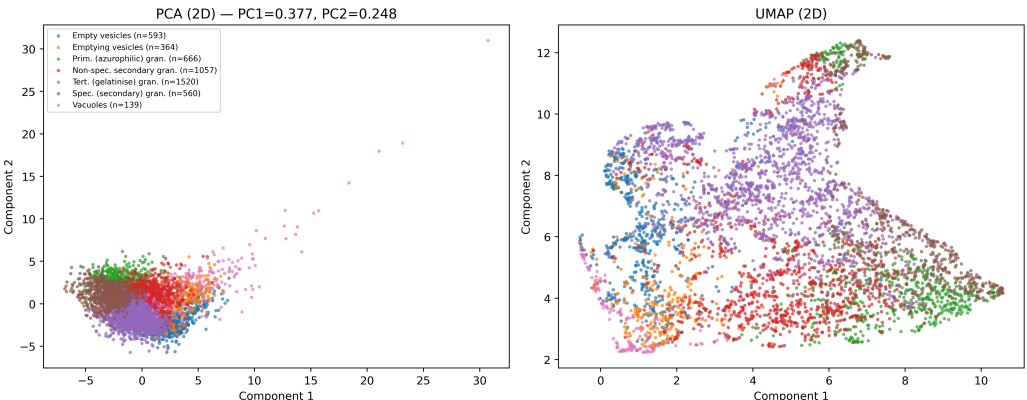

Figure 7: **Feature embeddings of neutrophil ultrastructures.** PCA (left, PC1 = 37.7%, PC2 = 24.8% variance) shows vacuoles and Tertiary (gelatinise) granules as partially distinct, while Primary (azurophilic) and Specific (secondary) granules overlap. UMAP (right) preserves local neighborhoods but reveals entanglement between vesicles and granules. These projections illustrate why hierarchical calibration with morphology–regularized CP is needed for reliable fine–grained classification.

CORRELATION ANALYSIS OF MORPHOLOGY AND INTENSITY FEATURES

Beyond the 2D projections in Fig. 7, we quantify the linear dependencies among geometry, intensity, and texture descriptors using Pearson correlations on the NUS–TEM training features. Table 5 reports representative feature pairs, both overall and aggregated across fine classes. Geometry features (e.g., area and equivalent diameter) show strong internal correlation, as expected, whereas geometry vs. intensity and intensity vs. texture exhibit only modest correlations, indicating that these groups provide complementary information for the classifier.

Table 5: Representative Pearson correlations between feature groups on NUS–TEM (train). $\rho_{\text{overall}}$ is computed over all instances; $\rho_{\text{min}}$ and $\rho_{\text{max}}$ denote the minimum and maximum per-class correlations, respectively, across the seven fine classes.

| Feature pair | $\rho_{\text{overall}}$ | $\rho_{\text{min}}$ | $\rho_{\text{max}}$ |
|---|---|---|---|
| Area vs. Eq. diameter | 0.82 | 0.71 | 0.89 |
| Area vs. Perimeter | 0.79 | 0.68 | 0.88 |
| Circularity vs. Eccentricity | -0.63 | -0.71 | -0.52 |
| Area vs. Mean intensity | 0.18 | 0.05 | 0.29 |
| IQR vs. Mean intensity | 0.24 | 0.11 | 0.36 |
| IQR vs. GLCM contrast | 0.31 | 0.19 | 0.42 |
| Mean intensity vs. GLCM homogeneity | -0.27 | -0.39 | -0.15 |

Overall, geometry features are internally coherent but only weakly to moderately correlated with intensity and texture cues. This pattern supports our design choice of combining morphology, intensity, and texture descriptors in the base classifier and in the morphology-regularized score used by MR–HCP.

**Feature projections (PCA/UMAP).** Figure 7 shows 2D PCA (left; PC1=37.7%, PC2=24.8%) and UMAP (right) embeddings of the 19D morphology+intensity features. Vacuoles cluster distinctly; Tertiary (gelatinise) granules are compact; Primary (azurophilic) and Specific (secondary) granules overlap; vesicles vs. granules exhibit local mixing. These patterns motivate the two-stage taxonomic gate and the morphology-regularized score used by MR–HCP.

GLOBAL COVERAGE–SIZE SWEEP VS. BASELINES (NUS–TEM)

Figure 8 extends the global comparison by sweeping the nominal target for all methods on NUS–TEM (GT crops). The left panel shows coverage vs. average set size: across the practically relevant

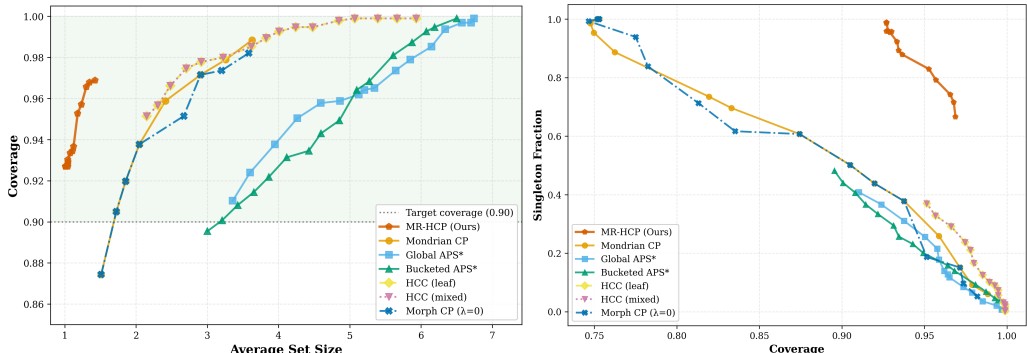

Figure 8: **Coverage–efficiency and singleton trade-offs on NUS–TEM (GT crops). Left:** Coverage vs. average set size across nominal targets for MR–HCP and all baselines (Mondrian CP, Global APS*, Bucketed APS*, HCC, Morph-CP); lower-left is better. **Right:** Singleton fraction vs. coverage for the same sweep, showing that MR–HCP maintains a high proportion of singletons at a given coverage level compared to the baselines.

range near 0.90–0.96 coverage, MR–HCP occupies the lower-left region of the plot, achieving comparable or higher coverage than Mondrian CP and Morph-CP at substantially smaller set sizes, and remaining far more efficient than APS variants and HCC, whose curves lie to the right (larger sets) for similar coverage. The right panel plots singleton fraction vs. coverage for the same sweep and confirms that, at a given coverage level, MR–HCP sustains a higher proportion of singleton predictions than all baselines, supporting the global trade-off reported in Table 1.

OPERATING-POINT SELECTION ON *calib_tune*

We select the MR–HCP operating point by sweeping $(\alpha_{\text{sup}}, \alpha_{\text{sub}})$ on *calib_tune* and choosing the Pareto solution that *minimizes* calibration average set size subject to achieving near-nominal calibration coverage. Figure 9 summarizes the sweep: panel **(a)** shows calibration coverage vs. $\alpha_{\text{sub}}$ with one curve per $\alpha_{\text{sup}}$, and panel **(b)** shows the corresponding calibration average set size. As expected, increasing $\alpha_{\text{sup}}$ uniformly relaxes coverage (vertical separation between curves), while increasing $\alpha_{\text{sub}}$ reduces set size with a mild coverage trade-off (movement along a curve). The selected operating point $(\alpha_{\text{sup}}, \alpha_{\text{sub}}) = (0.10, 0.05)$ lies near the knee of this surface, with calibration coverage $\approx 0.876$ and calibration average set size $\approx 1.09$, and achieves $\approx 0.937$ coverage with average set size $\approx 1.12$ on *TEST*; this is the operating point used throughout the main MR–HCP results, while nearby settings are reported for sensitivity in the appendix.

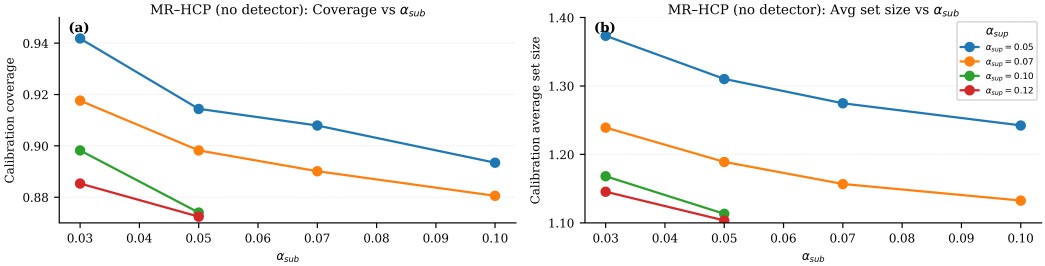

Figure 9: **Calibration sweep (MR–HCP, no detector). (a)** Calibration coverage vs. $\alpha_{\text{sub}}$, one curve per $\alpha_{\text{sup}}$. **(b)** Calibration average set size vs. $\alpha_{\text{sub}}$ (same color coding). The chosen operating point $(\alpha_{\text{sup}}, \alpha_{\text{sub}}) = (0.10, 0.05)$ is selected by minimizing set size subject to near-nominal calibration coverage and is used for the main NUS–TEM results.

IMAGE-CONDITIONAL CALIBRATION

We stratify by an image-level intensity surrogate $z$ into $K$ bins and estimate per-class thresholds within each bin. On *TEST*, panel **(c)** of Fig. 10 shows that global coverage remains essentially stable as $K$ increases (from 0.945 at $K=0$ to 0.947 at $K=4$), while the average set size changes only

slightly (from 1.161 to 1.170), indicating that image-conditional binning yields at most a modest efficiency adjustment without sacrificing validity. Panel **(a)** reports per-class coverage variability *across eligible bins* (we ignore bins with $< 5$ ground-truth samples). At $K=2$, several fine classes yield SD$= 0$ because only *one* bin meets the eligibility threshold; this does *not* imply intensity invariance, only insufficient support in the other bin. At $K=4$, more bins become eligible and SD turns non-zero, surfacing genuine between-bin variation for the classes most sensitive to intensity stratification.

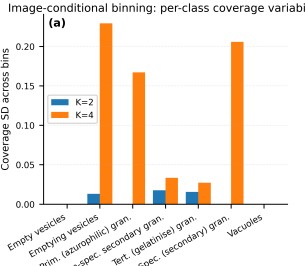 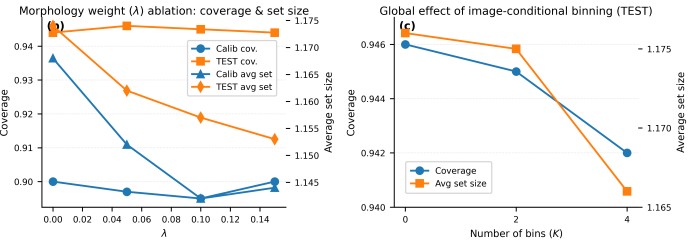

Figure 10: **Binning variability and morphology ablation.** **(a)** Per-class coverage variability (standard deviation) across image-conditional bins (K=2 vs. K=4), highlighting which fine classes are most sensitive to intensity stratification. SD$(K=2)=0$ for some classes because only one bin was eligible, not because coverage was identical across both bins. **(b)** Morphology weight ($\lambda$) ablation for both calibration and TEST. Coverage remains stable while sets shrink slightly up to $\lambda=0.15$. **(c)** Global effect of image-conditional binning on *TEST*: coverage stays essentially stable ($0.945 \rightarrow 0.947$) while sets change only slightly ($1.161 \rightarrow 1.170$) by $K=4$, indicating near-constant efficiency without sacrificing validity.

### MORPHOLOGY ABLATION ($\lambda$)

We vary the Stage 2 score $s_\lambda(x,y) = (1-\pi_y) + \lambda r_y$ and report calibration and TEST performance in Table 6. Morphology regularization slightly shrinks sets without harming coverage: on *TEST*, coverage remains in the narrow range $\approx 0.945$–$0.944$ while the average set size moves only slightly around $\approx 1.16$–$1.15$ as $\lambda$ increases to $0.15$ (see the table for exact values). The joint calibration/TEST behavior across $\lambda$ is shown in Fig. 10 b, where coverage (left axis) and average set size (right axis) evolve smoothly with $\lambda$.

Table 6: Ablation over morphology weight $\lambda$ (calibration $\rightarrow$ TEST).

| $\lambda$ | Calib cov. | Calib avg set | TEST cov. | TEST avg set |
|------|-----------|---------------|-----------|--------------|
| 0.00 | 0.905 | 1.173 | 0.945 | 1.158 |
| 0.05 | 0.905 | 1.170 | 0.945 | 1.149 |
| 0.10 | 0.903 | 1.170 | 0.945 | 1.154 |
| 0.15 | 0.908 | 1.168 | 0.944 | 1.153 |

### ABLATIONS ON SCORE DESIGN AND HIERARCHY

To justify the additive form of the morphology–regularized score, we ablate the contribution of hierarchy and morphology as well as alternative score combinations. Table 7 compares a hierarchy-only configuration ($\lambda = 0$), a morphology-aware multiplicative variant, a soft-clip variant, and the full MR–HCP. All variants achieve very similar coverage, but the additive MR–HCP form yields the best joint size–singleton tradeoff (smallest average set size with the highest singleton fraction), supporting the choice made in the main text.

As summarized in Table 7, hierarchy-only ($\lambda=0$), multiplicative, and soft-clip variants yield comparable coverage, but the full MR–HCP additive score attains the smallest average set size and highest singleton fraction. This confirms that morphology provides complementary information beyond the hierarchy alone and supports the additive design choice highlighted in Section 4.4.

Table 7: Ablations on score design and hierarchy on NUS–TEM (TEST, GT crops). All variants use the same $(\alpha_{\text{sup}}, \alpha_{\text{sub}})$; we report overall coverage, average set size, singleton fraction, and singleton accuracy.

| Variant | Coverage | Avg set size | Singleton frac. | Singleton acc. |
|---|---|---|---|---|
| Hierarchy-only ($\lambda = 0$) | 0.9366 | 1.1235 | 0.8786 | 0.9352 |
| Multiplicative | 0.9356 | 1.1160 | 0.8872 | 0.9322 |
| Soft-clip ($\tau = 0.5$) | 0.9345 | 1.1246 | 0.8786 | 0.9315 |
| MR–HCP (full additive) | 0.9356 | 1.1149 | 0.8883 | 0.9323 |

RELIABILITY AT THE SUPER-CLASS LEVEL

In Table 8 We report Expected Calibration Error (ECE) and Brier score for the four super-classes used in Stage 1 (lower is better). These metrics directly assess the calibrated probabilities that drive the Stage 1 gate, which is the most consequential decision in MR–HCP. Macro ECE is $\approx 0.038$ and macro Brier is $\approx 0.023$, indicating well-calibrated super probabilities that support valid gating.

Table 8: MR–HCP (no detector): super-class reliability on *TEST*.

| Super class | ECE $\downarrow$ | Brier $\downarrow$ |
|---|---|---|
| Light Granules | 0.0660 | 0.0441 |
| Dark Granules | 0.0460 | 0.0349 |
| Vesicles | 0.0388 | 0.0122 |
| Vacuoles | 0.0022 | 0.0008 |
| Macro (mean) | 0.0383 | 0.0230 |

# D  YOLO–MR–HCP: ADDITIONAL RESULTS

**Pipeline summary (self-contained).** We pair a 4-way YOLO segmenter for localization and super-class logits with a 7-way morphology-aware hierarchical conformal predictor (MR–HCP) that produces fine-level prediction sets. Calibration uses detections on `calib_core`; hyperparameters are chosen on `calib_tune`; evaluation is on TEST. Detections are associated to GT masks via Hungarian matching with IoU$\geq 0.5$. Unless stated, all metrics are computed on *matched* detections, and we report the matched fraction as a detector-recall proxy.

OPERATING POINTS UNDER DETECTOR COUPLING

**Overall trade-off on matched detections.** We compare three operating points on detections that match a GT instance (IoU$\geq$0.5) so localization does not confound results. Table 9 aggregates coverage, average set size ($|\Gamma|$), singleton rate, and the matched fraction; Fig. 11a plots their coverage–size positions. Three behaviors emerge: (i) *Without T* is the reference pack; (ii) *With $T^\star$* shifts probability mass at the super level while leaving conformal thresholds fixed, explaining the coverage drop at nearly fixed $|\Gamma|$; (iii) the *Post-tuning* pack refits thresholds for efficiency, moving left (smaller $|\Gamma|$) with a controlled coverage trade-off. Fig. 11b shows the class dependence of this trade-off: vesicle subtypes remain robust, whereas dark-granule subclasses are most sensitive to $T^\star$ and benefit from post-tuning.

**Matched vs. unmatched (clarification).** The matched fraction is identical across settings (Table 10) because MR–HCP alters only label sets on top of YOLO outputs and does not affect localization/association. Consequently, differences in Table 9 arise on the *matched* rows; for *unmatched* rows, the average set size is constant by design (a fixed fallback policy) and is included only to show that detector recall is orthogonal to the conformal stage.

**Interpreting *Post-tuning* vs. *Without T*.** The *Post-tuning* pack in Table 9 is intended as an illustrative operating point under imperfect detections, rather than a universally preferred configuration. It demonstrates the coverage–efficiency trade-off relative to the *Without T* pack: on matched detections, *Post-tuning* achieves approximately 22% smaller prediction sets (average $|\Gamma|$: $5.60 \rightarrow 4.37$) at the cost of roughly 12% lower coverage ($0.891 \rightarrow 0.782$). For applications that prioritize strict cov-

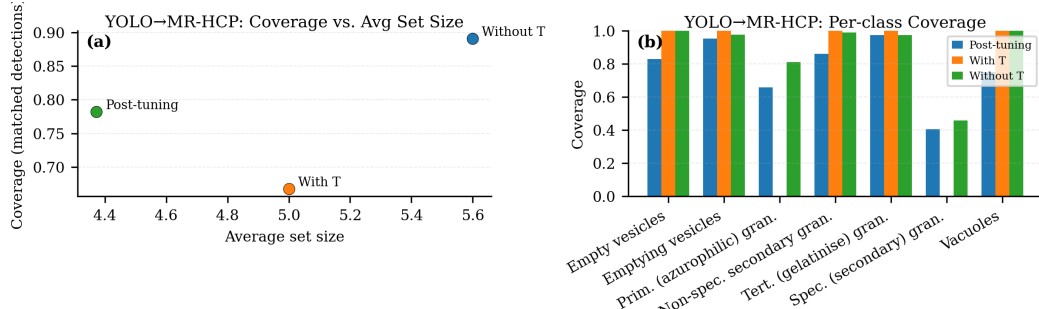

Figure 11: **YOLO→MR–HCP on matched detections (IoU≥0.5).** **(a)** Coverage vs. average set size for three operating points: *Without T*, *With $T^\star$* (post-hoc temperature scaling with fixed thresholds), and *Post-tuning* (retuned thresholds). The matched fraction is constant ($\approx 0.738$, $n=349$), so differences reflect label-set behavior rather than detector recall. **(b)** Per-class coverage (grouped bars) under the same settings; vesicles remain high, while dark-granule subclasses are most affected by $T^\star$ and improved by post-tuning.

Table 9: TEST (IoU$\geq 0.5$ matched) summary for YOLO→MR–HCP.

| Experiment | Coverage | Avg $|\Gamma|$ | Singleton | Matched frac | $n$ |
|---|---|---|---|---|---|
| Without T | 0.891 | 5.599 | 0.006 | 0.738 | 349 |
| With $T^\star$ | 0.668 | 5.000 | 0.000 | 0.738 | 349 |
| Post-tuning pack | 0.782 | 4.372 | 0.066 | 0.738 | 349 |

erage, *Without T* is therefore more appropriate, whereas *Post-tuning* may be preferable in workflows where ambiguous cases are routinely triaged by experts.

Table 10: Matched vs. unmatched breakdown on TEST.

| Experiment | Matched frac | Avg $|\Gamma|$ (matched) | Avg $|\Gamma|$ (unmatched) |
|---|---|---|---|
| Without T | 0.738 | 5.599 | 5.000 |
| With $T^\star$ | 0.738 | 5.000 | 5.000 |
| Post-tuning pack | 0.738 | 4.372 | 5.000 |

COVERAGE DECOMPOSITION: MATCHED VS. UNMATCHED DETECTIONS

To isolate the effect of detection errors on coverage, we decompose the YOLO→MR–HCP results into detections that match a ground-truth instance at IoU$\geq 0.5$ and unmatched detections. Table 11 shows that on matched detections (96% of test instances), MR–HCP attains high coverage with compact sets, whereas unmatched detections exhibit poor coverage despite similar set size.

These results support the interpretation given in the main text: the main coverage loss in the end-to-end pipeline arises from detector-induced distribution shift (missed or poorly localized objects), rather than from the conformalization mechanism itself.

SENSITIVITY TO IOU MATCHING THRESHOLD

Coverage declines as the matching threshold tightens—most noticeably between 0.5 and 0.7—while average set size is essentially unchanged (Table 12). This indicates that label-set efficiency is stable to the association rule; the main effect of stricter IoU is to exclude borderline matches rather than inflate $|\Gamma|$. All sweep points in Fig. 6 are evaluated at IoU$\geq 0.5$ for consistency.

MORPHOLOGY WEIGHT ($\lambda$) ABLATION

Holding $(\alpha_{\text{sup}}, \alpha_{\text{sub}})$ fixed, increasing $\lambda$ trades tiny coverage changes for modest set-size movement (Table 13). In practice, a small nonzero $\lambda$ pulls sets toward morphologically consistent labels without

Table 11: Coverage decomposition for YOLO→MR–HCP on NUS–TEM (TEST).

| Subset | Fraction of instances | Coverage | Avg set size |
|---|---|---|---|
| Matched | 0.96 | 0.9653 | 1.12 |
| Unmatched | 0.04 | 0.1892 | 1.13 |

Table 12: Coverage and size vs. IoU threshold (TEST, matched at the stated IoU).

| Experiment | IoU thr. | Coverage | Avg $|\Gamma|$ |
|---|---|---|---|
| Without T | 0.30 | 0.891 | 5.599 |
| Without T | 0.50 | 0.891 | 5.599 |
| Without T | 0.70 | 0.849 | 5.615 |
| With $T^\star$ | 0.30 | 0.668 | 5.000 |
| With $T^\star$ | 0.50 | 0.668 | 5.000 |
| With $T^\star$ | 0.70 | 0.624 | 5.000 |

degrading validity, which is why the post-tuning pack adopts $\lambda > 0$. The aggregate effect of $\lambda$ relative to $(\alpha_{\mathrm{sup}}, \alpha_{\mathrm{sub}})$ is also visible in the sweep cloud and the frontier in Fig. 6a–b.

Table 13: Ablation over morphology weight $\lambda$ (TEST, matched).

| $\lambda$ | Coverage | Avg $|\Gamma|$ |
|---|---|---|
| 0.00 | 0.900 | 5.676 |
| 0.10 | 0.888 | 5.533 |
| 0.15 | 0.885 | 5.530 |
| 0.20 | 0.882 | 5.544 |
| 0.25 | 0.880 | 5.630 |
| 0.30 | 0.880 | 5.702 |

CALIBRATION→TEST GENERALIZATION (CONCISE)

At the chosen post-tuning configuration, the calibration-tune snapshot achieved coverage 0.906 with $|\Gamma|$=4.807, and the final TEST evaluation achieved 0.782 with $|\Gamma|$=4.372 (Table 9). The shift reflects distribution differences between *calib_tune* and TEST and conditioning on matched detections at evaluation time. The selected point lies near the Pareto frontier in Fig. 6b, prioritizing smaller sets while keeping coverage competitive.

IMAGE-CONDITIONAL BINNING (CONCISE)

Adding image-conditional bins for Stage-2 thresholds slightly improves robustness to intensity shifts with only a marginal effect on average set size. Binning lets each image sub-population use thresholds estimated from similar images, smoothing between-image calibration drift. In our matched-detection evaluation this yields a small coverage gain at near-constant set size and is orthogonal to the operating-point choices in Fig. 6a–b. (For MR–HCP without the detector, the analogous effect and per-class variance diagnostics appear in Fig. 10a,c.)

WHY POST-HOC TEMPERATURE SCALING ($T^\star$) CAN HURT

$T^\star$ is learned on the *super* level and then applied *after* conformal thresholds $(\hat{q}_S, \hat{q}_F)$ are fixed. This reshapes super-class probabilities without re-estimating the quantiles that define the sets, creating a mismatch: nonconformity scores change but thresholds do not, so coverage can drop while average set size remains essentially unchanged. This explains the "With $T^\star$" behavior relative to "Without T" in Table 9 and its location in Fig. 11.

# E IMPLEMENTATION AND ENVIRONMENT (CONCISE)

## E.1 SPLITS (IMAGE-WISE)

All partitions are *image-wise* to prevent crop leakage across splits. Calibration detections are divided into *calib_core* (quantiles) and *calib_tune* (hyperparameters); TEST is disjoint. Group-aware splitting uses image_id. Counts are shown in Table 14.

Table 14: Image-wise splits and annotation counts.

| Subset | #Images | #Annotations | Usage |
|--------|---------|--------------|-------|
| train | 25 | 3349 | Fit base classifier |
| calib | 8 | 619 | Conformal calibration |
| test | 5 | 931 | Final evaluation |

## E.2 BASE CLASSIFIER

A feature-based multiclass LightGBM is trained with class weights inversely proportional to class frequency. Features are standardized using *train*-only statistics. Hyperparameters are listed in Table 15.

Table 15: LightGBM hyperparameters.

| Parameter | Value | Parameter | Value | Parameter | Value |
|-----------|-------|-----------|-------|-----------|-------|
| boosting_type | gbdt | num_leaves | 64 | max_depth | -1 |
| learning_rate | 0.05 | n_estimators | 1000 | subsample | 0.8 |
| colsample_bytree | 0.8 | reg_alpha | 0.1 | reg_lambda | 0.1 |
| class_weight | balanced | | | | |

## E.3 PROBABILITY CALIBRATION

Multinomial temperature scaling is fitted on *calib_tune* to obtain $T^\star$; calibrated probabilities $\tilde{\pi}$ are used throughout. When $T^\star$ is active, conformal thresholds are estimated *after* calibration to keep scores and quantiles aligned.

## E.4 MR–HCP STAGES AND SCORE CONSTRUCTION

Stage 1 aggregates super-class probabilities $\pi_g^S = \sum_{y \in \mathcal{Y}_g} \tilde{\pi}_y$ and applies class-conditional split-CP to obtain $\{\hat{q}_g^S\}$ on *calib_core*. Stage 2 refines within retained supers using

$$s_\lambda(x, y) = (1 - \tilde{\pi}_y(x)) + \lambda\, r_y(x),$$

where $r_y$ is a clipped Mahalanobis penalty built on *calib_core* (95th-percentile scaling). $(\alpha_{\text{sup}}, \alpha_{\text{sub}})$ and $\lambda$ are selected on *calib_tune*. Quantile ties use fixed randomized smoothing. Empty buckets/bins fall back to global thresholds (see §C).

Table 16: MR–HCP stages and calibration sources.

| Stage | Inputs | Calibration source |
|-------|--------|--------------------|
| Stage-1 (Super) | $\pi_g^S$ from $\tilde{\pi}$ | $\{\hat{q}_g^S\}$ on calib_core |
| Stage-2 (Fine) | $s_\lambda(x, y)$ within retained $g$ | $\{\hat{q}_{y\vert g}\}$ on calib_core |
| Morph penalty | $\phi(x); d_y^2; r_y = \min(1, d_y^2/c_y)$ | $c_y$ from calib_core |
| $\lambda$ tuning | Coverage–size trade-off | calib_tune (disjoint) |
| Output | $\Gamma_\alpha(x)$ | coverage $\geq 1 - \alpha_{\text{sup}} - \alpha_{\text{sub}}$ |

## E.5 FEATURE SET

Hand-crafted descriptors include geometry (area, perimeter, equivalent diameter), shape (circularity, solidity, eccentricity), intensity (quartiles $Q_1, Q_2, Q_3$, mean, skewness, IQR), and texture (GLCM contrast, homogeneity). Standardization uses *train* statistics; rare missing values are imputed with *train* medians.

### E.6 Raabin–WBC Dataset: Additional Details

The Raabin–WBC dataset used in our experiments focused on five fine classes: *Neutrophil*, *Lymphocyte*, *Eosinophil*, *Monocyte*, and *Basophil*. We adopt an image-wise split into *train*, *calib*, and *test* to avoid crop leakage, mirroring the protocol used for NUS–TEM.

Table 17 summarizes the split sizes, and Table 18 reports the per-class distribution across splits. The resulting dataset contains 14,514 annotated cells, with class imbalance (particularly for Basophils) handled via class weights in the base LightGBM classifier.

Table 17: Raabin–WBC: image-wise splits and annotation counts.

| Subset | #Images | #Annotations | Usage |
|--------|---------|--------------|-------|
| train | — | 10,159 | Fit base classifier |
| calib | — | 2,177 | Conformal calibration (core+tune) |
| test | — | 2,178 | Final evaluation |

Table 18: Raabin–WBC: per-class distribution across splits.

| Class | train | calib | test |
|-------|-------|-------|------|
| Neutrophil | 6223 | 1334 | 1334 |
| Lymphocyte | 2422 | 519 | 520 |
| Eosinophil | 746 | 160 | 160 |
| Monocyte | 557 | 119 | 119 |
| Basophil | 211 | 45 | 45 |

This split yields enough calibration support in each class for reliable split-conformal quantile estimation and downstream MR–HCP calibration.

### E.7 Reproducibility and Artifacts

We fix seeds for splitting and model training; quantile ties use fixed randomized smoothing. We store: the $(\alpha_{\text{sup}}, \alpha_{\text{sub}})$ sweep and chosen point (Fig. 9); $\lambda$ ablation and image-conditional binning diagnostics (Fig. 10); reliability (ECE/Brier) at super and fine levels; the `pack` JSON (thresholds, hyperparameters); and TEST prediction-set CSVs with `image_id`, `ann_id`, `y_true`, $\Gamma$, and `set_size`.

#### YOLO detector (for the YOLO→MR–HCP pipeline)

We use a segmentation-capable YOLO variant to localize instances and produce super-class logits. Training follows the same image-wise split as the classifier. Inputs are letterboxed to a fixed resolution; data augmentation and optimization follow the trainer defaults unless stated in `hyp.yaml`. The best checkpoint is selected by *validation* mAP@50 (mask); inference uses fixed confidence and NMS thresholds.

Detections are associated to GT masks by Hungarian matching with mask IoU$\geq 0.5$, and all YOLO→MR–HCP results are reported on the matched subset (matched fraction reported alongside results). The detector does not alter conformal thresholds; MR–HCP operates on its outputs only.

## F Algorithmic Summary and Usage Notes

This section links the four pseudocode listings and clarifies how they are used together in practice.

**How the pieces fit.** Algorithms 1–4 cover two deployment contexts: *(i) MR–HCP without a detector*, where instances are pre-segmented crops (Algorithms 1 and 2), and *(ii) YOLO→MR–HCP*, where MR–HCP operates on YOLO detections (Algorithms 3 and 4). In both contexts, calibration is performed once (on *calib_core* with tuning on *calib_tune*) to produce fixed thresholds and hyperparameters; inference then applies those thresholds to new samples. The image-conditional option (§3.5) is orthogonal: if enabled, bin-specific quantiles are learned during calibration and selected at inference via the image statistic $z$.

**Guarantees (shared).** All four algorithms implement split-conformal prediction with hierarchical gating. With $(\eta, \lambda)$ fixed on a disjoint tuning slice, and under exchangeability (globally or within image bins), the resulting prediction sets achieve finite-sample coverage at least $1 - \alpha_{\text{sup}} - \alpha_{\text{sub}}$. Randomized ties at the quantile boundary are handled via the standard coin-flip rule.

**What to run when.**

- **Crops-only setting (no detector).** Run Alg. 1 once to produce $\{\hat{q}_g^S\}$, $\{\hat{q}_{y|g}\}$ (or binned versions), $(\mu_y, \Sigma_y, c_y)$, and $\lambda^\star$; then apply Alg. 2 for each test instance.
- **Detection-aware setting (YOLO→MR–HCP).** Run Alg. 3 once on matched detections to obtain fused super thresholds, fine thresholds, morphology stats, and $\lambda^\star$; then apply Alg. 4 to each detection at test time.

**One-liners for common choices.**

- **Fusion weight $\eta$.** If you do not fuse detector and classifier at the super level, set *(detector-only)* $\eta{=}1$ or *(classifier-only aggregate)* $\eta{=}0$. In the crops-only setting, $\eta$ is not used.
- **Image-conditional binning $K$.** If you do not use image-conditional calibration, set $K{=}0$ (algorithms fall back to global quantiles). If enabled, choose $K \in \{2, 4, \dots\}$ and route samples by $z$.
- **Morphology penalty $\lambda$.** Use a small nonzero $\lambda^\star$ selected on *calib_tune* to reduce set size without harming coverage; set $\lambda{=}0$ to recover standard HCP.
- **Tie handling.** Keep the default 0.5 tie-break to preserve exact finite-sample validity with discrete scores.

**Mapping to figures.** The offline/online workflow in Fig. 2 corresponds to *calibration* (top lane; Algorithms 1 or 3) and *inference* (bottom lane; Algorithms 2 or 4). For detector coupling, the "matched detections" protocol (§D) aligns with Alg. 3.

---

**Algorithm 1:** MR–HCP (no detector): Calibration

---

**Input:** Calibration set $(x_i, y_i) \in \mathbb{R}^d \times \mathcal{Y}$ split into *calib_core* and *calib_tune*;
   hierarchy $\{\mathcal{Y}_g\}_{g \in \mathcal{G}}$; error levels $(\alpha_{\text{sup}}, \alpha_{\text{sub}})$; bins $K \in \{0, 2, 4, \dots\}$
**Output:** Super thresholds $\{\hat{q}_g^S\}$; fine thresholds $\{\hat{q}_{y|g}\}$ (or per-bin $\{\hat{q}_{y|g,k}\}$);
   morphology stats $\{(\mu_y, \Sigma_y, c_y)\}$; tuned $\lambda^\star$ (from *calib_tune*)
**1. Probability calibration.** Fit temperature $T^\star$ on *calib_tune*; obtain calibrated probs $\tilde{\pi}_y(x)$.
**2. Super nonconformity.** For each $(x, y)$ in *calib_core* and each $g \in \mathcal{G}$, define
   $s_{\text{sup}}(x, g) \leftarrow 1 - \sum_{u \in \mathcal{Y}_g} \tilde{\pi}_u(x)$.
**3. Super thresholds.** For each $g$, collect $\{s_{\text{sup}}(x_i, g)\}$ over *calib_core* with true super $g^\star$ and
   set $\hat{q}_g^S \leftarrow \text{SplitQuant}(\{s_{\text{sup}}(x_i, g^\star{=}g)\}, \alpha_{\text{sup}})$.
**4. Morphology statistics.** On *calib_core*, compute $\mu_y \leftarrow \mathbb{E}[\phi(x) \mid y]$, $\Sigma_y$ (regularized); for each
   sample compute $d_y^2(x) = (\phi(x) - \mu_y)^\top \Sigma_y^{-1} (\phi(x) - \mu_y)$ and set $c_y \leftarrow$ 95th percentile of $d_y^2$.
**5. Candidate $\lambda$ grid.** On *calib_tune*, for each $\lambda \in \Lambda$ form
   $s_\lambda(x, y) \leftarrow (1 - \tilde{\pi}_y(x)) + \lambda \cdot \min(1, d_y^2(x)/c_y)$.
**6. Fine thresholds (no binning).** If $K{=}0$, for each $g$ and $y \in \mathcal{Y}_g$ set
   $\hat{q}_{y|g} \leftarrow \text{SplitQuant}(\{s_\lambda(x_i, y_i) \ : \ y_i{=}y\}, \alpha_{\text{sub}})$ on *calib_core*.
**7. Fine thresholds (binned).** If $K{>}0$, compute bin edges on *calib_core* via quantiles of an
   image statistic $z$; for each bin $k$, class $y$ set
   $\hat{q}_{y|g,k} \leftarrow \text{SplitQuant}(\{s_\lambda(x_i, y_i) \ : \ y_i{=}y, \ x_i \in B_k\}, \alpha_{\text{sub}})$ with fallback to global $\hat{q}_{y|g}$ if a
   bin is sparse.
**8. Select $\lambda^\star$.** Choose $\lambda$ on *calib_tune* that optimizes coverage–set-size; fix $\lambda^\star$.

---

---

**Algorithm 2:** MR–HCP (no detector): Inference

---

**Input:** Test instance $x$, calibrated probs $\tilde{\pi}(x)$; $\{\hat{q}_g^S\}$; $\{\hat{q}_{y|g}\}$ or $\{\hat{q}_{y|g,k}\}$; $(\mu_y, \Sigma_y, c_y)$; $\lambda^\star$; $K$

**Output:** Prediction set $\Gamma(x) \subseteq \mathcal{Y}$

**1. Super gate.** For each $g \in \mathcal{G}$, set $s_{\sup}(x, g) \leftarrow 1 - \sum_{u \in \mathcal{Y}_g} \tilde{\pi}_u(x)$ and include $g$ in $\mathcal{G}_\alpha(x)$ iff IncludeTie$(s_{\sup}(x, g), \hat{q}_g^S) = 1$.

**2. If** $\mathcal{G}_\alpha(x) = \varnothing$, force $g^\star \leftarrow \arg\max_g \sum_{u \in \mathcal{Y}_g} \tilde{\pi}_u(x)$ and set $\mathcal{G}_\alpha(x) \leftarrow \{g^\star\}$.

**3. Fine stage.** For each $g \in \mathcal{G}_\alpha(x)$ and $y \in \mathcal{Y}_g$: compute $r_y(x) \leftarrow \min(1, d_y^2(x)/c_y)$ with $d_y^2(x) = (\phi(x) - \mu_y)^\top \Sigma_y^{-1} (\phi(x) - \mu_y)$;
  set $s_{\lambda^\star}(x, y) \leftarrow (1 - \tilde{\pi}_y(x)) + \lambda^\star r_y(x)$;
  if $K = 0$, include $y$ iff IncludeTie$(s_{\lambda^\star}(x, y), \hat{q}_{y|g}) = 1$;
  if $K > 0$, route $x$ to bin $k$ by $z(x)$ and include $y$ iff IncludeTie$(s_{\lambda^\star}(x, y), \hat{q}_{y|g,k}) = 1$.

**4. Fallback.** If no $y$ was included for some retained $g$, add $\arg\max_{y \in \mathcal{Y}_g} \tilde{\pi}_y(x)$.

**5. Return** $\Gamma(x)$ (union over retained supers).

---

**Algorithm 3:** YOLO→MR–HCP: Detection-aware Calibration

---

**Input:** Calibration images with GT masks/labels; YOLO detector/segmenter; feature extractor $\phi(\cdot)$; fine classifier for $\tilde{\pi}(\cdot)$; IoU threshold $\tau$; fusion weight $\eta \in [0, 1]$ (fixed on tuning); $(\alpha_{\sup}, \alpha_{\sub})$; bins $K$

**Output:** Super thresholds $\{\hat{q}_g^S\}$ for fused scores; fine thresholds $\{\hat{q}_{y|g}\}$ or $\{\hat{q}_{y|g,k}\}$; $(\mu_y, \Sigma_y, c_y)$; $\lambda^\star$

**1. Run YOLO on calibration images.** Collect detections $b$ (boxes/masks) with coarse logits; associate to GT via Hungarian matching at IoU$\geq \tau$.

**2. Build detection tuples.** For each matched detection $b$: extract $\phi(b)$ from the YOLO mask; compute $\tilde{\pi}_y(b)$ from the fine classifier; record true labels $(g^{\text{true}}, y^{\text{true}})$.

**3. Fused super probabilities.** For each detection $b$ and super $g$, set
$\bar{\pi}_g^S(b) \leftarrow \eta \, \pi_g^{\text{YOLO}}(b) + (1 - \eta) \sum_{u \in \mathcal{Y}_g} \tilde{\pi}_u(b)$,
$s_{\sup}(b, g) \leftarrow 1 - \bar{\pi}_g^S(b)$.

**4. Super thresholds.** For each $g$, set $\hat{q}_g^S \leftarrow$ SplitQuant$(\{s_{\sup}(b, g^{\text{true}} = g)\}, \alpha_{\sup})$ using matched detections only.

**5. Morphology statistics.** Compute $(\mu_y, \Sigma_y, c_y)$ from $\phi(b)$ as in Alg. 1 (on matched detections).

**6. $\lambda$ tuning and fine thresholds.** Proceed as in Alg. 1 Steps 5–8 but using matched detections, yielding $\lambda^\star$ and $\{\hat{q}_{y|g}\}$ (or binned $\{\hat{q}_{y|g,k}\}$).

---

**Algorithm 4:** YOLO→MR–HCP: Inference on a Detection

---

**Input:** Detection $b$ with YOLO mask/logits; fine classifier; $(\hat{q}_g^S)$; $(\hat{q}_{y|g})$ or $(\hat{q}_{y|g,k})$; $(\mu_y, \Sigma_y, c_y)$; $\lambda^\star$; $\eta$; $K$

**Output:** Prediction set $\Gamma(b) \subseteq \mathcal{Y}$

**1. Probs and features.** Obtain $\pi^{\text{YOLO}}(b)$; extract $\phi(b)$; compute $\tilde{\pi}(b)$.

**2. Fused super score.** For each $g$, $\bar{\pi}_g^S(b) \leftarrow \eta \, \pi_g^{\text{YOLO}}(b) + (1 - \eta) \sum_{u \in \mathcal{Y}_g} \tilde{\pi}_u(b)$; set $s_{\sup}(b, g) \leftarrow 1 - \bar{\pi}_g^S(b)$.

**3. Super gate.** Keep $g$ in $\mathcal{G}_\alpha(b)$ iff IncludeTie$(s_{\sup}(b, g), \hat{q}_g^S) = 1$; if empty, keep $g^\star \leftarrow \arg\max_g \bar{\pi}_g^S(b)$.

**4. Fine stage.** For each kept $g$ and $y \in \mathcal{Y}_g$, compute $r_y(b) \leftarrow \min(1, d_y^2(b)/c_y)$ and $s_{\lambda^\star}(b, y) \leftarrow (1 - \tilde{\pi}_y(b)) + \lambda^\star r_y(b)$.
Use global $\hat{q}_{y|g}$ if $K = 0$; otherwise route to bin $k$ by $z(b)$ and compare to $\hat{q}_{y|g,k}$.

**5. Fallback.** If no $y$ included within some retained $g$, add $\arg\max_{y \in \mathcal{Y}_g} \tilde{\pi}_y(b)$.

**6. Return** $\Gamma(b)$.

---

# G TERMINOLOGY

APS VARIANTS

We report *APS (Global)* and *APS (Bucketed)*. APS constructs a set by ordering labels by $\tilde{\pi}_y$ and including them until a calibrated cumulative mass exceeds a threshold. "*Bucketed*" applies the same procedure with image-conditional calibration (NUM_BINS $> 0$): calibration is performed within $z$-stratified image bins and bin-specific thresholds are used at test time.

