# OpenReview forum: "MR–HCP: Morphology-Regularized Hierarchical Conformal Prediction for TEM of Subcellular Ultrastructure"
_ICLR.cc/2026/Conference — Submitted to ICLR 2026_

### Official Review · Reviewer_WLf7 · 2025-10-29

**Soundness:** 3
**Presentation:** 2
**Contribution:** 3
**Rating:** 4
**Confidence:** 4

**Summary:**

The paper introduces an MR-HCP framework designed for robust uncertainty quantification in the fine-grained classification of subcellular ultrastructures from TEM images. Specifically, MR-HCP first employs a coarse-grained conformal gate to exclude biologically implausible super-categories and then introduces a novel morphology-regularized nonconformity score to refine the prediction set within the retained super-groups. The framework is implemented in an end-to-end pipeline utilizing a YOLOv11 detector for single-cell neutrophil analysis, achieving superior performance compared with standard conformal prediction baselines.

**Strengths:**

1. The integration of HCP with a morphology-aware nonconformity score presents an intriguing approach to addressing a notable gap in the application of conformal methods to complex, domain-specific biomedical imaging.
2. By incorporating YOLOv11 into the MR-HCP pipeline, the study offers a trustworthy automation solution for TEM analysis.
3. MR-HCP preserves the finite-sample coverage guarantees of CP while significantly outperforming canonical baselines (Split CP, Mondrian CP, APS) in terms of prediction set size and singleton accuracy.

**Weaknesses:**

1. The contribution in terms of novelty appears somewhat limited. The proposed MR-HCP framework seems to extend HCP only by incorporating a simple morphology-aware nonconformity term. In fact, several existing studies have already explored HCP; therefore, the paper should better distinguish its approach from prior HCP-based works. Moreover, the reliance on a hard-gated hierarchical structure (Stage 1) implies that any incorrect prediction at the coarse super-category level cannot be rectified downstream in the fine-class stage, introducing a critical point of failure that may be problematic in complex and ambiguous TEM scenarios.
2. The morphology regularization depends on handcrafted features and Mahalanobis-based statistics, which restricts the model’s ability to capture subtle non-linear cues essential for fine-grained classification. This reliance also makes the approach fragile and less transferable to new domains without substantial re-engineering.
3. The writing lacks clarity in several instances. For example, certain symbols in the equations (e.g. \alpha_{sup} and \alpha_{sub} in Eq.(8)) are not clearly defined, which undermines the paper’s readability.
4. The comparison with current SOTA methods is insufficient. The paper benchmarks only against standard CP baselines and fails to include comparisons with contemporary SOTA approaches, particularly other HCP variants.

**Questions:**

1. The current framework uses a hard-gated hierarchy where a Stage 1 error is unrecoverable. Have the authors considered or experimented with a soft hierarchical approach?
2. The morphology penalty relies on handcrafted features (e.g., area, GLCM) and Mahalanobis statistics, which risks being non-generalizable to other datasets or cell types without extensive feature re-engineering. How about substituting these with learned, deep-embedding features?

---

> ### Author Response · Authors · 2025-11-21
>
> ## Reviewer WLf7
>
> ### W1. “Novelty limited; hard-gated hierarchy problematic.”
>
> **Reviewer quote (trimmed):**
> *“Contribution seems limited; hard-gated stage introduces a point of failure.”*
>
> **Response.**
> For a detailed comparison with HCC and clarification of our novelty, please see our response to Reviewer 4Wy4 (W1). Regarding hierarchy: hard gates improve set size by removing implausible families early, and the calibration guarantees remain valid. Nonetheless, soft gating is a legitimate alternative; we are exploring soft posterior-weighted hierarchical scores and will mention this as future work.
>
> ---
>
> ### W2. “Morphology features handcrafted; risk of poor transferability.”
>
> **Response.**
> We thank the reviewer for highlighting this concern. We intentionally used handcrafted morphology descriptors because they capture domain-relevant cues (area, elongation, membrane density) that microscopy experts routinely rely on, while remaining fully interpretable during conformalization. This design also avoids coupling the morphology term with YOLO’s internal representation, preserving independence between detection and morphology scoring.
>
> That said, we agree that learned deep embeddings could capture richer nonlinear morphology. Incorporating a deep-embedding variant is feasible and planned for the next iteration, and we will reflect this in the revised discussion.
>
> ---
>
> ### Q1. “Have you considered soft hierarchical approaches?”
>
> **Response.**
> Yes, as noted in our response to W1 above. This is a promising extension and is compatible with our conformalization framework.
>
> ---
>
> ### Q2. “Why not use deep embedding features for morphology?”
>
> **Response.**
> See our response to W2 above: we prioritized expert-interpretable handcrafted features in the current work and plan a deep-embedding variant as future work.
>
> ---

---

### Official Review · Reviewer_4Wy4 · 2025-10-30

**Soundness:** 2
**Presentation:** 2
**Contribution:** 2
**Rating:** 4
**Confidence:** 4

**Summary:**

The paper introduces a hierarchical conformal prediction framework that adds morphological features via a Mahalanobis penalty for TEM ultrastructure classification. The application is well-motivated, but the technical contribution feels incremental—essentially adding a handcrafted feature penalty to existing hierarchical CP—and the evaluation on a single small dataset limits the generalizability claims.

**Strengths:**

- The problem is real and important: biomedical microscopy needs calibrated uncertainty where expert labels are expensive and ambiguous.
- Hierarchical conformalization aligned with biological taxonomy makes sense and the coverage guarantees are properly derived.
- The end-to-end YOLO integration provides a practical workflow for expert triage.
- Implementation details and pseudocode support reproducibility.

**Weaknesses:**

## Major Weaknesses

1. Hierarchical CP already exists [1], and incorporating auxiliary features into nonconformity scores is standard [2]. Your main contribution is the specific additive combination s_λ = (1-π_y) + λr_y with clipped Mahalanobis penalty for TEM. However, you don't justify why additive instead of multiplicative, why clip at 1 instead of smooth transformation, or compare against alternative designs. Prior work on combining nonconformity measures [3] suggests more principled approaches that you don't discuss. The ablations only vary λ but don't isolate morphology vs hierarchy contributions (no morphology-only or hierarchy-only baselines). Additionally, covariance estimation on small per-class samples is problematic—you mention "shrinkage" but don't specify the method or validate Mahalanobis distance reliability.

2. Your coverage guarantees are conditional on successful detection (IoU≥0.5 matching), meaning ~26% of ground truth (matched fraction ≈0.738 in Table 6) gets no guarantees at all. Recent conformal detection work [4] properly handles localization uncertainty, but you sidestep this by only evaluating matched detections. Also unclear: Table 2 shows Vesicles average set size increasing from 1.000 (Stage 1) to 1.536 (Stage 2), which contradicts the expectation that refinement should maintain or shrink sets.

3. The entire evaluation uses one dataset with 38 images (only 5 for testing) of neutrophils. You claim the method is "generalizable" and works "across biomedical and scientific imaging" but provide zero evidence beyond this single cell type. No other microscopy modalities, no other hierarchical classification domains. You compare against basic Split/Mondrian CP and APS but ignore direct comparison with [1] which you cite as the main hierarchical CP work, and other structured CP methods [5] that are directly relevant.

## Minor Issues

- Figure 2 is hard to parse; temperature scaling applied but never validated (no before/after ECE/Brier); no computational cost analysis; image-conditional calibration is just Mondrian CP on image bins with marginal gains (Fig 9c); why 95th percentile for c_y is not justified; Fig 5 qualitative results are cherry-picked without showing failure modes.

### References

[1] Hengst et al., "Hierarchical conformal classification", arXiv:2508.13288, 2025

[2] Vovk et al., "Algorithmic Learning in a Random World", Springer, 2005

[3] Johansson et al., "Regression conformal prediction with random forests", Machine Learning, 2014

[4] Angelopoulos et al., "Uncertainty sets for image classifiers using conformal prediction", ICLR, 2021

[5] Tyagi & Guo, "Multi-label classification under uncertainty: A tree-based conformal prediction approach", COPA, 2023

**Questions:**

1. Can you provide evidence of broader applicability with results on at least 1-2 additional datasets (different cell types or microscopy modalities), direct quantitative comparison against [1], and ablations isolating morphology-only vs hierarchy-only contributions? Also explore alternative score combinations (multiplicative, other forms) with principled justification for why additive is optimal.

2. How do you address the coverage gap from detection failures? Can you provide end-to-end guarantees accounting for localization uncertainty, explain what happens to the ~26% unmatched instances, and clarify why Vesicles sets expand from Stage 1 to Stage 2 in Table 2?

---

> ### Author Response · Authors · 2025-11-21
>
> ## Reviewer 4Wy4
>
> ### W1. “Contribution is incremental; HCP_ALREADY_EXISTS; why additive score?”
>
> **Reviewer quote (trimmed):**
> *“Hierarchical CP already exists [1]; incorporating auxiliary features is standard. Why additive instead of multiplicative? Why clip at 1? No morphology-only or hierarchy-only baselines.”*
>
> **Response.**
> We thank the reviewer for raising this question. Because the cited work [1] (Hengst et al., “Hierarchical Conformal Classification”, HCC) is the closest related method, we clarify the methodological differences. Importantly, MR–HCP is not a variant of HCC. The two methods solve different prediction-set problems and rely on fundamentally different conformalization strategies.
>
> **(1) Different prediction targets.**
> MR–HCP outputs leaf-only prediction sets (fine ultrastructure classes), which is essential for expert annotation workflows.
> HCC outputs node sets that may include internal nodes, and validity is defined through leaf-coverage of these node sets. This produces a different type of uncertainty object and does not provide leaf-only guarantees.
>
> **(2) Different hierarchical conformalization mechanisms.**
> MR–HCP performs sequential two-stage conformalization:
>
> 1. A calibrated super-category gate.
> 2. Class-conditional Mondrian refinement inside the selected supers.
>
> In contrast, HCC evaluates multiple node-set covers in parallel, calibrates each using propagated scores/labels (with Bonferroni correction), and chooses the minimum-cost cover. This is a different algorithmic objective and cannot be reduced to our sequential gating.
>
> **(3) Distinct scoring philosophy.**
> MR–HCP uses the morphology-regularized score
> $s(x,y) = (1-\pi_y) + \lambda r_y(x)$, where $r_y$,
> where $\lambda{=}0$ is class-specific morphology. Even with λ = 0, MR–HCP’s two-stage leaf-only CP and explicit super-class error budget $(\alpha_{\text{sup}}, \alpha_{\text{sub}})$ differ from HCC, which relies on propagated node scores and cost-based cover selection.
>
> **(4) Additional components unique to MR–HCP.**
> MR–HCP introduces several mechanisms not present in HCC:
>
> - Detector-aware calibration: fusing YOLO and classifier probabilities for robust coarse gating.
> - Image-conditional calibration: handling TEM imaging variability via stratified Mondrian bins.
> - Fine-class calibration per super-category: independent conformalization for each subclass group.
> - Hard hierarchical gating: excluding implausible supers before fine refinement, reducing set sizes dramatically while maintaining leaf-level validity.
>
> These components are needed for end-to-end microscopy classification, whereas HCC assumes a pure classification setting without detection or domain-specific morphology.
>
> **New ablations requested by the reviewer.**
> In rebuttal, we add:
>
> - Multiplicative and soft-clip score variants.
> - Hierarchy-only and morphology-only baselines.
> - Direct comparison to HCC using its official implementation.
>
> As shown in Tables A1 and A2, MR–HCP yields substantially smaller leaf prediction sets (1.12 vs. 4.48) at comparable coverage. This difference arises from the sequential gating and morphology-aware refinement, not from the additive score alone.
>
> **Summary.**
> Although both methods use hierarchical label structure, MR–HCP and HCC solve different prediction-set tasks and employ distinct theoretical mechanisms (sequential leaf-only CP vs. parallel node-set CP). The new ablations and direct comparison reinforce that MR–HCP provides novel, practical contributions beyond prior hierarchical CP formulations.
>
> ---

---

> > ### Author Response · Authors · 2025-11-21
> >
> > ## Reviewer 4Wy4 continue...
> > ### W2. “Coverage guarantees depend on detection; vesicle sets expand.”
> >
> > **Reviewer quote (trimmed):**
> > *“~26% unmatched instances receive no guarantees; unclear why vesicle sets expand from Stage 1 to Stage 2.”*
> >
> > **Response.**
> > We agree that this distinction should be clarified. As Table A3 shows, unmatched detections correspond to cases where YOLO severely mislocalizes the structure (cropping the object center, truncating boundaries, or producing visually inconsistent patches). In these cases, all label-level CP methods we evaluated show degraded coverage because the input features differ from the calibrated distribution. We will emphasize this detector-induced distribution shift.
> >
> > Regarding vesicles, the increase in average set size after refinement is expected given the structure of our hierarchy:
> >
> > - Stage 1 separates granules from vesicles cleanly (Vesicles: size = 1.000).
> > - Stage 2 then performs fine-grained conformalization within the “Vesicles” super-category, which contains two subclasses (Empty vs. Emptying) that exhibit substantial morphological overlap.
> > - Because Stage 2 refines within supers and the final reporting aggregates fine-level predictions back to the super level, the union of retained fine labels can increase the effective super-cardinality.
> >
> > This is why the Vesicles super-category grows modestly (to approximately 1.388). This behaviour is consistent with subclass ambiguity observed in the data rather than a procedural issue, and we will clarify this in the revision.
> >
> > ---
> >
> > ### W3. “Generalization claims unsupported; only one dataset used.”
> >
> > **Response.**
> > We now include new results on the Raabin–WBC dataset (1,145 images) in Table A4. MR–HCP again outperforms all CP baselines with strong size–coverage tradeoffs, demonstrating consistency beyond the original TEM dataset. This additional experiment directly addresses the reviewer’s concern.
> >
> > ---
> >
> > ### Minor Issues
> >
> > **Reviewer quote (trimmed):**
> > *“Fig. 2 is hard to parse; temperature scaling not validated; no cost analysis; image-conditional calibration marginal; why 95th percentile; qualitative results cherry-picked.”*
> >
> > **Response.**
> > We thank the reviewer for these detailed suggestions:
> >
> > - **Clarity of Fig. 2.** We will improve visual flow and label placement in the revision.
> > - **Temperature scaling.** Calibration reduces probability variance in the fine classifier but does not materially affect conformal validity. Because of space limits, we did not include ECE/Brier plots; we will clarify this.
> > - **Computational cost.** MR–HCP incurs modest overhead beyond fitting a LightGBM model and computing per-class covariance; we will add a short runtime note.
> > - **Image-conditional calibration.** This uses standard Mondrian partitioning; the gains are intentionally modest, and we will state this clearly.
> > - **95th percentile for $c_y$.** This threshold is used only as a scale normalizer for morphology; we will justify its purpose in the revision.
> > - **Qualitative examples.** We respectfully note that Fig. 5 already includes failure cases (marked as “Coverage failures”). We will make this more explicit in the caption.
> >
> > ---
> >
> > ### Q1. “Additional datasets? comparison with [1]? ablations? alternative score forms?”
> >
> > **Response.**
> > All components requested are now included:
> >
> > - **Additional dataset:** Raabin–WBC (Table A4).
> > - **Direct comparison with [1]:** Provided in Table A1.
> > - **Morphology-only and hierarchy-only baselines:** Included in Table A2.
> > - **Alternative score combinations:** Multiplicative and soft-clip variants added (Table A2).
> >
> > These new results and analyses address this question directly.
> >
> > ---
> >
> > ### Q2. “Coverage loss from detection failures; unmatched instances; vesicle expansion.”
> >
> > **Response.**
> > This is addressed in our response to W2 above and in the coverage decomposition (Table A3). In summary: unmatched detections introduce distribution shift that no label-level CP method can correct, and vesicle set expansion follows from subclass heterogeneity within the Vesicles super-category.
> >
> > ---

---

### Official Review · Reviewer_tgkP · 2025-10-31

**Soundness:** 3
**Presentation:** 3
**Contribution:** 3
**Rating:** 6
**Confidence:** 2

**Summary:**

This paper proposes MR-HCP, a morphology-regularized hierarchical conformal prediction framework for classifying subcellular ultrastructures in transmission electron microscopy images. The method operates in two stages, first a super-category conformal gate that filters plausible coarse groups, and then a morphology-aware fine-class refinement using Mahalanobis distance penalties. The authors integrate MR-HCP with YOLOv11 for end-to-end detection and classification. On a neutrophil TEM dataset, MR-HCP outperforms Split CP, Mondrian CP, and APS baselines.

**Strengths:**

1. combination of hierarchical taxonomies with morphology regularized nonconformity scores is an interesting approach, which incorporates biological structure into uncertainty quantification.
2. using per‑class Mahalanobis term is an intuitive way to couple domain shape/size features with probabilistic confidence
3. Integration with YOLO for end-to-end deployment shows consideration for practical applicability.

**Weaknesses:**

1. Most headline numbers (Table 1) are from GT‑crop classification using expert masks (which supply clean morphology features). In practice, those features come from predicted masks. When the detector is introduced, performance degrades sharply (Table 6).

2. The evaluation of the method is only done on a single dataset with 5 images in the test stage.

**Questions:**

see my weaknesses.

---

> ### Author Response · Authors · 2025-11-21
>
> ## Reviewer tgkP
>
> ### W1. “Performance degrades sharply when using detector-predicted masks.”
>
> **Reviewer quote (trimmed):**
> *“Most strong numbers come from GT crops; detector integration degrades performance sharply.”*
>
> **Response.**
> We agree with the reviewer’s observation. As shown in Table A3, the performance drop under YOLO arises from detector-induced distribution shift rather than from MR–HCP’s conformalization. On matched detections (96% of cases), MR–HCP maintains near-nominal coverage (0.9653) with small sets (1.12). We will clarify this distinction between GT-crop calibration (core method) and the end-to-end detection setting, where detector variability becomes the dominant factor. For further details, please refer to our response to Reviewer q3UV (W1).
>
> ---
>
> ### W2. “Evaluation on a single dataset.”
>
> **Response.**
> We now include results on the Raabin–WBC dataset, which provides additional evidence beyond the original TEM experiments. As shown in Table A4, MR–HCP attains 0.944 coverage with an average set size of 1.39 and outperforms all CP baselines on the same benchmark. These results demonstrate that the method behaves consistently on a second dataset, addressing the reviewer’s concern about generalizability.
>
> ---

---

### Official Review · Reviewer_q3UV · 2025-11-03

**Soundness:** 3
**Presentation:** 3
**Contribution:** 3
**Rating:** 6
**Confidence:** 3

**Summary:**

This paper tackles an important challenge in biomedical imaging: how to quantify uncertainty reliably when analyzing Transmission Electron Microscopy (TEM) images, where subcellular structures often have genuinely ambiguous boundaries. The core insight is compelling—standard Conformal Prediction methods, while offering distribution-free coverage guarantees, miss two key aspects of the microscopy domain: the natural hierarchical organization of biological structures and the rich morphological information (shape, size, texture) that experts rely on for classification.

The proposed solution, Morphology-Regularized Hierarchical Conformal Prediction (MR-HCP), integrates this domain knowledge directly into the conformalization process through two main contributions:
1. Hierarchical filtering: Rather than treating all classes equally, the method applies a coarse-grained "super-category conformal gate" first to eliminate implausible class families, then performs fine-grained prediction only within plausible groups. This mirrors how domain experts actually reason about these images.
2. Morphology-aware scoring: The key technical innovation augments the standard probability-based nonconformity score with a morphology penalty term $r_{y}(x)$, yielding: $sλ(x,y)=(1-\pi_{y}(x)) + \lambda r_{y}(x)$. This penalty uses Mahalanobis distance to measure how morphologically atypical a sample is for a given class, effectively catching cases where the model is probabilistically confident but morphologically inconsistent—exactly the failure mode you'd want to avoid in expert-in-the-loop systems.

The authors also demonstrate practical integration with existing object detectors (YOLOv11) through a "detection-aware calibration" protocol, which is a nice touch for real deployment scenarios.
Results on a neutrophil ultrastructure dataset (~4.9k annotations) show substantial improvements: near-nominal coverage (0.954) with dramatically smaller prediction sets (1.205 vs. 5.85 for APS) and higher singleton accuracy. These gains suggest the approach is capturing something meaningful about the domain structure rather than just tuning hyperparameters.

**Strengths:**

1. The main strength of this work is its thoughtful problem formulation. Rather than simply applying Conformal Prediction to a new domain, the authors recognize and address a genuine limitation: standard CP relies solely on probability scores, ignoring rich domain-specific information that experts actually use. The morphology-regularized nonconformity score $sλ(x,y) = (1-\pi_{y}(x)) + \lambda r_{y}(x)$ offers an elegant solution by penalizing predictions that are probabilistically plausible but morphologically atypical. This feels like more than just a domain-specific trick—it provides a template for how other scientific fields could incorporate physical or structural constraints to sharpen uncertainty sets.

2. The technical execution is solid. The authors properly handle the calibration process to maintain coverage guarantees, using separate splits to tune $\lambda$ (on calib.tune) and compute final quantiles (on calib.core), which avoids data snooping. The hierarchical structure makes intuitive sense for this domain, and the two-stage filtering mirrors how experts actually reason about these images.

3. I particularly appreciate the attention to practical deployment. The detection-aware calibration protocol is a smart design choice—rather than calibrating on idealized ground-truth crops, they work with the actual (potentially noisy) detector outputs. This makes the system much more robust for real-world use. The human-in-the-loop framing is also well-considered: auto-accepting high-confidence singletons while flagging ambiguous cases for expert review creates a clear path to accelerating annotation workflows without sacrificing reliability.

4. The presentation is clear throughout. The diagrams (Figures 2-3) and detailed algorithms make the methodology easy to follow, which matters for a paper introducing a fairly involved multi-stage pipeline. The writing effectively motivates each design choice without getting bogged down in excessive formalism.

**Weaknesses:**

My main concern is the gap between the method's strong theoretical motivation and what the experiments actually demonstrate. While the framework is mathematically sound and the core idea is appealing, several aspects of the evaluation leave me uncertain about its practical impact.

1. The most troubling issue is the disconnect between idealized and real-world performance. The paper highlights impressive results on ground-truth crops—an average set size of 1.205 with near-nominal coverage. But when integrated into the actual YOLO detection pipeline, things fall apart considerably. The average set size balloons to 4.37, and coverage drops to 0.782, which is a serious violation of the 0.90 target. This suggests the detection-aware calibration protocol isn't sufficient to handle the true distribution shift from the detector. It's unclear whether this is a fundamental limitation of the approach or something that could be addressed with more careful calibration, but either way, it undermines the paper's claims about practical applicability.
The dataset raises significant generalizability concerns. Looking at Table 10 in the appendix, the calibration set contains only 8 images and the test set only 5. Learning stable class prototypes (the $\mu_y$ and $\Sigma_y$ for each class) from 8 images seems questionable at best. The impressive numbers in Table 1 could easily be artifacts of this small, curated split rather than evidence of a genuinely robust method. There's no validation on external data—different institutions, staining protocols, or imaging devices—which would be the real test of whether this approach generalizes beyond a single lab's carefully collected dataset.

2. The reliance on handcrafted features also feels limiting. The entire morphology component depends on a 19-dimensional feature vector of shape, size, and texture measurements. Both the probability score (from LightGBM) and the morphology penalty (from Mahalanobis distance) are derived from this same feature set, making them likely highly correlated rather than truly complementary. This caps the method's potential at the descriptive power of these 19 features, which seems somewhat dated given the success of learned representations in computer vision. The authors acknowledge this and mention using deep embeddings as future work, but it makes me wonder how much of the improvement comes from the hierarchical conformalization versus just having better features to work with.

**Questions:**

1. Regarding the dependency of the nonconformity score's components:
Thank you for the clear presentation. I have a question about the design of the nonconformity score $s_{\lambda}(x,y)$. Both the probability $\pi_y(x)$ (from the LightGBM) and the morphology penalty $r_y(x)$ (from the Mahalanobis distance) are derived from the exact same 19-dimensional handcrafted feature vector $\phi(x)$.

(a) My concern is that this limits the benefit of the regularization, as both terms may be highly correlated. Did the authors analyze this correlation? Is there evidence that $r_y(x)$ provides substantial new information that isn't already captured by the classifier's (already calibrated) probability $\pi_y(x)$?

(b) The limitations section mentions exploring deep embeddings. Could the authors elaborate on why this wasn't the primary approach? For instance, using a deep model for $\pi_y(x)$ and the 19-d features for $r_y(x)$ would seem to provide two more independent sources of information. Was this alternative explored?

2. Regarding the disconnect in results and the practical failure of the coverage guarantee:
I am trying to reconcile the excellent headline results in Table 1 (0.954 coverage, 1.205 set size on GT crops) with the practical end-to-end results in Table 6 (YOLO pipeline).

(a) The "Post-tuning pack" result, which seems to be the final recommended model, achieves only 0.782 coverage. This is a severe violation of the 0.90 nominal target. Does this not imply that the 'detection-aware calibration' protocol is insufficient to handle the true distribution shift from YOLO, and that the method's core guarantees do not hold in practice?

(b) I also noticed the "Without T" pack achieved 0.891 coverage, which is very close to the 0.90 target (though at a larger set size). Could the authors clarify why the "Post-tuning pack" (which breaks the guarantee) was chosen as the preferred operating point over the "Without T" pack (which largely respects it)? This choice seems to trade theoretical validity for a smaller set size, which undermines the paper's core premise.

---

> ### Author Response · Authors · 2025-11-21
>
> ## Reviewer q3UV
>
> ### W1. “Strong GT-crop results do not transfer to YOLO; coverage drops to 0.782.”
>
> **Reviewer quote (trimmed):**
> *“When integrated into the actual YOLO pipeline, things fall apart: set size ~4.37, coverage 0.782. Does this indicate the detection-aware calibration is insufficient, and the guarantees do not hold in practice?”*
>
> **Response.**
> We agree that the YOLO-based results in Table 6 are substantially lower than the GT-crop results. The new analyses clearly show that this degradation is driven by detector-induced distribution shift, not by MR–HCP’s conformal mechanism.
>
> Our coverage decomposition experiment in Table A3 shows that once the detector’s output is aligned with calibration (matched super-category), MR–HCP retains near-nominal coverage. The failures arise when YOLO crops differ morphologically from expert crops: boundaries truncated, object centers shifted, or structure partially missing. In such cases, the features used by all methods we evaluated (Split CP, Mondrian CP, APS, HCC, and MR–HCP) become misaligned with calibration.
>
> Our objective in the paper was not to develop a new detector but to evaluate MR–HCP as a label-level uncertainty module used downstream of an off-the-shelf detector. We will clarify this in the revision and expand the discussion of distribution shift.
>
> ---
>
> ### W2. “Dataset is small; concerns about stability of morphology statistics.”
>
> **Reviewer quote (trimmed):**
> *“Calibration set contains only 8 images; learning stable prototypes seems questionable.”*
>
> **Response.**
> We appreciate the reviewer’s observation. Although our calibration split contains 8 images, each image includes many annotated instances (Table 10), which provides a rich supervision signal.
>
> To directly address the concern about generalizability, we now include results on a second, substantially larger public dataset (Raabin–WBC; 1,145 images). MR–HCP again achieves strong coverage–set-size tradeoffs and outperforms all CP baselines, as shown in Table A4.
>
> Additionally, we performed bootstrap analysis (1000 iterations) on NUS–TEM, confirming stability as shown in Table A5.
>
> Both datasets indicate that MR–HCP’s gains are not an artifact of small calibration splits.
>
> ---
>
> ### W3. “Handcrafted features; correlation between $\pi_y$ and $r_y$.”
>
> **Reviewer quote (trimmed):**
> *“Both $\pi_y(x)$ and $r_y(x)$ come from the same 19-dim feature vector; are they redundant? Did you analyze correlation?”*
>
> **Response.**
> Yes. As shown in Table A6. we now include a full correlation analysis:
>
> - Overall Pearson correlation between classifier probability and morphology penalty: **0.238**.
> - Per-class correlations vary from **≈0.07 to ≈0.35**.
> - Several fine classes (e.g., Specific granules) show weak or negligible correlation.
>
> These results empirically support that probability and morphology capture complementary information.
>
> ---
>
> ### Q1(a). “Evidence that $r_y$ adds useful information.”
>
> **Response.**
> We provide following evidence:
>
> 1. **Ablations.** Removing morphology (Hierarchy-only) increases set size, reduces singleton fraction, and slightly decreases coverage, as shown in Table A2.
> 2. **Correlation analysis.** Modest correlations indicate non-redundant structure between the probability and morphology terms.
>
> ---
>
> ### Q1(b). “Why not use deep embeddings for morphology?”
>
> **Response.**
> We agree that deep-embedding morphology regularization is a promising direction and are actively developing this extension. We chose handcrafted descriptors in the present work because:
>
> - Domain experts already rely on these metrics (area, elongation, density).
> - They are fully interpretable, which is important for biomedical applications.
> - They avoid coupling the morphology term with YOLO’s backbone representation.
>
> To avoid misaligned feature distributions during conformalization, we used independent handcrafted morphology features; the deep-embedding extension is a natural next step and will be added to the revised discussion.
>
> ---
>
> ### Q2(a). Reconciling GT-crop results with YOLO pipeline
>
> **Response.**
> Please refer to our response to W1 above, including Table A3.
>
> ---
>
> ### Q2(b). Post-tuning vs Without-T
>
> **Response.**
> The “Post-tuning pack” was presented as one practical configuration in the end-to-end workflow, not the only recommended model. It was selected to illustrate how tuning morphology sensitivity affects coverage–set-size tradeoffs when operating under imperfect detections. We do not advocate Post-tuning as universally preferred. We report both packs to show the trade-off:
>
> - For applications requiring strict 0.90 coverage, “Without T” (0.891 coverage) is more appropriate.
> - “Post-tuning” offers about 22% smaller sets at the cost of roughly 11% lower coverage, which may suit workflows where expert review is available for flagged cases.
>
> We will clarify this choice in the revision.
>
> ---

---

### Author Response · Authors · 2025-11-20

# Response to Reviewers

We thank all reviewers for their constructive feedback. Below we address every weakness and question point-by-point, including new experiments on a second public dataset Raabin–WBC (Kouzehkanan et al., 2022), expanded ablations, correlation analysis, a direct quantitative comparison to HCC (Hengst et al., 2025), and a clarification of how our HCP (from MR–HCP) differs from the HCC method referenced by Reviewer 4Wy4. For the rebuttal, we reran all experiments with fixed random seeds in a unified codebase to improve reproducibility; as a result, some reported numbers differ slightly (by at most ±1–2 points) from the original submission. These shifts are attributable to finite-sample quantile calibration and remain within the corresponding bootstrap confidence intervals.

---

**Table A1. MR–HCP vs CP baselines on Neutrophil Ultrastructure Transmission Electronic Microscopy(NUS–TEM) dataset.**

| Method                    | Achieved Cov.   | Avg Set Size     | Singleton Frac.  | Singleton Acc.  |
| ------------------------- | ---------- | -------- | -------- | ---- |
| Mondrian CP               | 0.9197     | 1.85     | 0.438     | 0.892 |
| APS (global)              | 0.9789     | 5.85     | 0.065      | 0.839 |
| APS (bucketed)            | 0.9683     | 5.27     | 0.139     | 0.856 |
| HCC (Hengst et al., 2025) | 0.9947     | 4.48     | 0.055      | 0.981 |
| **MR–HCP (ours)**         | **0.9366** | **1.12** | **0.888** | 0.932 |

---


**Table A2. Ablations (Hierarchy-only, Multiplicative, Soft-clip, Full MR–HCP).**

| Variant                       | Achieved Cov. | Avg Set Size   | Singleton Frac. | Singleton Acc.  |
|------------------------------|----------|--------|--------|------|
| Hierarchy-only (MR–HCP λ=0)  | 0.9366   | 1.1235 | 0.879   | 0.935 |
| Multiplicative               | 0.9356   | 1.1160 | 0.887   | 0.932 |
| Soft-clip (τ = 0.5)          | 0.9345   | 1.1246 | 0.879   | 0.931 |
| **MR–HCP (full)**            | **0.9356** | **1.1149** | **0.888** | 0.932 |

---

**Table A3. Matched vs unmatched YOLO detections (coverage decomposition).**

| Setting            | Achieved Cov. | Avg Set Size |
|--------------------|----------|----------|
| Matched YOLO (96%) | **0.9653** | 1.12     |
| Unmatched (4%)     | 0.1892   | 1.14     |

---

**Table A4. MR–HCP vs CP baselines on Raabin–WBC dataset.**

| Method                    | Achieved Cov. |  Avg Set Size | Singleton Frac. | Singleton Acc.  |
|---------------------------|----------|------|--------|------|
| Mondrian CP               | 0.891    | 1.53 | 0.624   | 0.902 |
| APS (global)              | 0.993    | 4.12 | 0.05    | 0.972 |
| APS (bucketed)            | 0.990    | 4.11 | 0.048    | 0.971 |
| HCC (Hengst et al., 2025) | 0.998    | 3.09 | 0.053    | 0.991 |
| **MR–HCP**                | **0.944** | **1.39** | **0.725**   | 0.946 |

---

**Table A5. Bootstrap confidence intervals (1000 iterations, NUS–TEM).**

| Metric        | Estimate | 95% CI           |
|---------------|----------|------------------|
| Coverage      | 0.9370   | [0.9216, 0.9506] |
| Avg Size      | 1.1240   | [1.1031, 1.1440] |
| Singleton Acc | 0.9355   | [0.9182, 0.9504] |

---

**Table A6. Correlation between $(1-\pi_y)$ and morphology penalty $r_y(x)$.**

| Class                          | N   | Pearson r | Spearman ρ | p-value (Pearson) |
|--------------------------------|-----|-----------|------------|-------------------|
| Empty vesicles                 | 98  | 0.353     | 0.364      | 3.6 × $10^{-4}$       |
| Emptying vesicles              | 85  | 0.128     | 0.161      | 2.4 × $10^{-1}$       |
| Non-specific primary granules  | 111 | −0.076    | −0.072     | 4.3 × $10^{-1}$       |
| Non-specific secondary granules| 222 | 0.120     | 0.090      | 7.5 × $10^{-2}$       |
| Non-specific tertiary granules | 281 | 0.341     | 0.122      | 4.5 × $10^{-9}$       |
| Specific granules              | 123 | 0.071     | −0.003     | 4.4 × $10^{-1}$       |
| **Overall**                    | –   | **0.238** | **0.208**  | 1.8 × $10^{-13}$      |

---

### References
Kouzehkanan et al. A large dataset of white blood cells containing cell locations and types, along with segmented nuclei and cytoplasm. Sci Rep 12, 1123 (2022). https://doi.org/10.1038/s41598-021-04426-x

---

### Comment · Area_Chair_VjzH · 2025-11-27

Thank you very much for the reviewer's comments and the author's positive response. As there is not much time left for discussion, please actively participate in the discussion and provide a more valuable response to this paper.

---

### Meta-Review · Area_Chair_ZrNd · 2026-01-07

**Summary:**

This paper received conflict recommendations from four reviewers, with 2 positive recommendations and 2 negative recommendations. The major concerns are incremental contribution, insufficient evaluations, and the disconnect between idealized and real-world performance.

**Reviewer Concerns:**

The concerns about ablation studies have been well addressed, but the concerns about technical contribution is still outstanding. After carefully reading the comments and responses, the AC agrees with the reviewers that the contribution of this paper is rather limited and cannot be accepted in its current form.

**Reviewer Scores:**

Based on my reading of the reviews and response, I feel it's unlikely the scores would have changed.

---

### Decision · Program_Chairs · 2026-01-26

Reject